# Stimulus vignetting and orientation selectivity in human visual cortex

**Zvi N Roth[1]\*, David J Heeger[2,3], Elisha P Merriam[1]**

[1]Laboratory of Brain and Cognition, National Institute of Mental Health, National Institutes of Health, Bethesda, United States; [2]Department of Psychology, New York University, New York, United States; [3]Center for Neural Science, New York University, New York, United States

**Abstract** Neural selectivity to orientation is one of the simplest and most thoroughly-studied cortical sensory features. Here, we show that a large body of research that purported to measure orientation tuning may have in fact been inadvertently measuring sensitivity to second-order changes in luminance, a phenomenon we term 'vignetting'. Using a computational model of neural responses in primary visual cortex (V1), we demonstrate the impact of vignetting on simulated V1 responses. We then used the model to generate a set of predictions, which we confirmed with functional MRI experiments in human observers. Our results demonstrate that stimulus vignetting can wholly determine the orientation selectivity of responses in visual cortex measured at a macroscopic scale, and suggest a reinterpretation of a well-established literature on orientation processing in visual cortex.

DOI: https://doi.org/10.7554/eLife.37241.001

## Introduction

Primary visual cortex (V1) is likely the best studied sensory cortical area, and is a model for understanding broad principles of cortical processing. Orientation in V1 is one of the simplest and best studied cortical sensory features. Orientation is used as a model for understanding more complex feature processing in other cortical areas, and oriented V1-like receptive fields play an important role in successful computational models of vision. Yet the map of orientation in V1 is inadequately understood. At a fine scale, the map shows an orderly periodic structure with pinwheels in hypercolumns, which have a periodicity of about 1 mm in monkeys (*Hubel and Wiesel, 1963*; *Blasdel and Salama, 1986*; *Grinvald et al., 1986*; *Das and Gilbert, 1997*; *Maldonado et al., 1997*; *Ohki et al., 2006*), and about 2 mm in humans (*Adams et al., 2007*; *Yacoub et al., 2008*). While the fine-scale structure has been studied extensively, the coarse-scale structure of orientation selectivity is poorly established.

We and others, using fMRI, have described a coarse-scale orientation bias in human V1; each voxel exhibits an orientation preference that depends on the region of space that it represents (*Furmanski and Engel, 2000*; *Sasaki et al., 2006*; *Mannion et al., 2010*; *Freeman et al., 2011*; *Freeman et al., 2013*; *Larsson et al., 2017*). We observed a radial bias most clearly in the peripheral representation of V1: voxels that responded to peripheral locations near the vertical meridian tended to respond most strongly to vertical orientations; voxels along the peripheral horizontal meridian responded most strongly to horizontal orientations; likewise for oblique orientations. This phenomenon had gone mostly unnoticed in single-unit recording studies. fMRI is well-suited to measuring coarse-scale orientation biases because fMRI covers the entire retinotopic map in visual cortex.

The observation of a coarse-scale orientation bias raises two questions. First, what is the relative contribution of coarse- and fine-scale biases to fMRI measurements of orientation selectivity? A

\*For correspondence:
zvi.roth@mail.huji.ac.il

**Competing interests:** The authors declare that no competing interests exist.

number of previous studies have asserted that orientation preferences in fMRI measurements arise primarily from random spatial irregularities in the fine-scale columnar architecture (*Boynton, 2005*; *Haynes and Rees, 2005*; *Kamitani and Tong, 2005*). This conjecture has inspired a large literature on the use of multivariate statistical analyses to exploit these small biases, both to study orientation selectivity in V1 and to study a wide range of other functions throughout the brain. On the other hand, we have argued that the coarse-scale orientation bias is the predominant orientation-selective signal measured with fMRI, and that multivariate decoding analysis methods are successful because of it (*Freeman et al., 2011*, *2013*). The relative contribution of coarse- and fine-scale biases remains hotly debated (*Pratte et al., 2016*; *Alink et al., 2017*).

Second, while the neural architecture that gives rise to orientation selectivity is well understood, the neural basis for the coarse-scale orientation bias is unknown. *Carlson (2014)* proposed that the coarse-scale bias is a byproduct of the edge of the stimulus, referring to it as an 'edge effect.' By this account, the coarse-scale bias does not directly reflect enhanced neural responses for particular orientations, but rather arises from properties of the stimulus.

Here, we develop a theoretical account of the edge effect that *Carlson (2014)* described. We then tested this account empirically. We show that this effect applies not only to stimulus edges but to a much broader class of stimuli. We use the term 'stimulus vignetting,' rather than 'edge effect,' to emphasize that it is not the edge per se, but rather an interaction between the orientation of the stimulus and a second feature of the display (the aperture or 'vignette') that bounds the stimulus. While there are potentially multiple distinct neural mechanisms contributing to coarse-scale orientation biases, we focus here only on characterizing the contribution of stimulus vignetting. We used an image-computable model of V1 activity to generate predictions, which we then tested with fMRI experiments. Our results provide a framework for reinterpreting a wide-range of findings in the visual system.

## Results

### Coarse-scale bias and stimulus vignetting: Model simulations

We studied the impact of stimulus vignetting on coarse-scale orientation preference maps. The principle underlying stimulus vignetting is that the vignette spreads Fourier energy. The direction and amplitude of this spread is determined by the underlying stimulus spectrum and the aperture geometry, in much the same way that the Fourier transform of a sinusoid windowed by a rectangle consists of the sinusoid frequency convolved with a sinc function. Due to this spectral spread at the aperture edge, neurons with receptive fields straddling the aperture edge respond to a combination of spatial frequency and orientation components, different from neurons with receptive fields far from the edge (*Figure 1*).

We began by simulating the results of our previous fMRI experiment (*Freeman et al., 2011*) with an image-computable model of V1 (*Simoncelli et al., 1992*) (see Materials and methods: Theoretical Model). The inputs to the model were identical to the stimuli used by *Freeman et al. (2011)*. Specifically, the stimuli consisted of 16 evenly spaced orientation angles between 0° and 180°. All stimuli were vignetted by a 4° annulus (inner radius, 5°; outer radius, 9°). Both the inner and outer edges of the annulus were blurred with a 1° raised-cosine transition (centered on the inner and outer edges). We measured the model's responses to each image separately. Each pixel of the model's output can be thought of as a simulated neuron in the retinotopic map of V1. To simulate an fMRI voxel's response to the stimuli used in Freeman et al., we summed the model responses across all orientation channels. This procedure yielded a spatial array of simulated voxel responses with different population receptive fields (pRFs). The response of a single simulated fMRI voxel corresponded to a single spatial location on the map of neural responses (e.g., with a pRF straddling the aperture edge). This yielded a set of 16 simulated voxel maps, one for each stimulus orientation (*Figure 2*). Importantly, we summed across all of the orientation channels. It might be expected, therefore, that the output of the model would not exhibit any orientation tuning. But the simulated fMRI responses do in fact exhibit a clear coarse-scale bias for orientation (*Figure 2*), replicating previous results (*Carlson, 2014*). The coarse-scale orientation bias matched *Freeman et al. (2011)* in that the largest responses were observed for radial orientations (*Figure 2*).

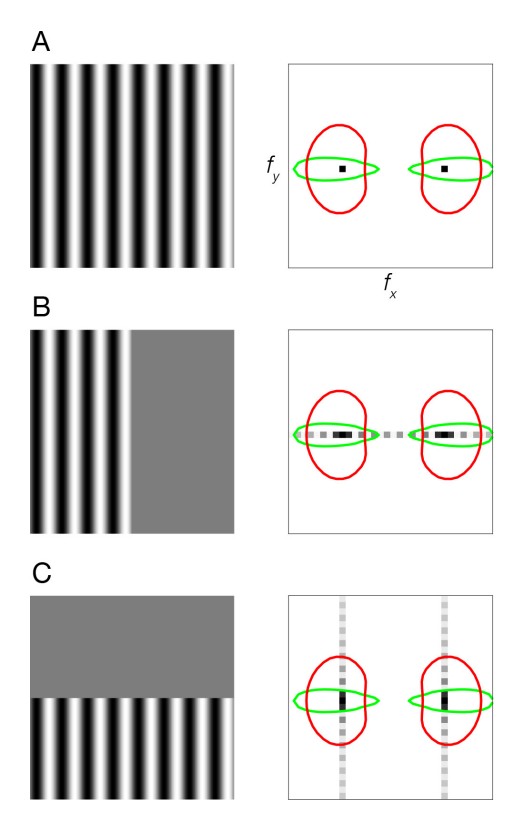

**Figure 1.** Stimulus vignetting. (**A**) Stimulus image (left) and Fourier amplitude (right) for sinusoidal grating that is infinite in extent. Black dots, Fourier amplitude restricted to a single spatial frequency and orientation component. Colored ovals, orientation tuning (polar angle cross-section through the oval) and spatial frequency tuning (radial cross-section) of two hypothetical V1 neurons. Red oval, neuron with broad orientation tuning and narrow spatial frequency tuning. Green oval, neuron with narrow orientation tuning and broad spatial frequency tuning. (**B**) Sinusoidal grating vignetted by vertical aperture. Light gray dots, spread of Fourier energy because of the aperture. (**C**) Sinusoidal grating vignetted by horizontal aperture.
DOI: https://doi.org/10.7554/eLife.37241.002

Having established that stimulus vignetting can produce a radial coarse-scale bias, we next asked whether the computational model predicts the influence of vignetting on responses to novel stimuli. The simulation results suggest that orientation-selective responses measured with fMRI could reflect, at least in part, the shape of the vignette, rather than the underlying orientation tuning of individual neurons. To test this possibility, we generated a novel set of stimuli with vignettes that were orthogonal to one another. If the vignette affects orientation bias, changing the vignette should have a predictable effect on orientation bias. The novel stimuli for this simulation were created as follows. Vertical and horizontal Cartesian gratings (the carrier) were multiplied by a radial or an angular polar grating (the modulator). These compound stimuli were then passed through the model. Subtracting horizontal from vertical carrier grating outputs resulted in an image of orientation bias for the model.

The model predicted that the coarse-scale orientation bias should depend on the modulator. The radial modulator evoked a coarse-scale bias that was radial (*Figure 3*). In other words, at regions around the vertical meridian there is a greater response for the vertical orientation than for the horizontal orientation (light regions in *Figure 3*), and vice versa around the horizontal meridian (dark regions in *Figure 3*). The model predicted that changing the orientation of the modulator should affect the orientation preference to the carrier grating. We found that the angular modulator evoked a coarse-scale bias that was tangential. Thus, at the vertical meridian the horizontal grating yielded higher energy, and at the horizontal meridian the vertical grating evokes higher energy responses.

The stimuli used in *Freeman et al. (2011)*, and in our simulations in *Figure 2*, had vignettes with gradual contrast changes, indicating that the impact of vignetting on orientation bias is not dependent on the presence of sharp edges. But is it possible that a more gradual edge would have mitigated vignetting effects? We answered this question by using a sinusoidal modulation that lacked any edges (*Figure 3*, left). We found similar results for both sinusoidal modulators and modulators with hard edges (i.e. square-wave), as expected, because vignetting arises from any contrast change in the image, rather than an 'edge' per se (see Discussion).

## Coarse-scale bias and stimulus vignetting: fMRI experiments

We found evidence of stimulus vignetting at both 3T and 7T field strengths, at different spatial resolutions, and for both square wave and sinusoidal modulators, as predicted by the theoretical model. The radial modulator evoked a radial bias, that is, orientation preferences pointing inward toward the fovea (*Figure 4*, top row). The angular modulator evoked a tangential bias, that is, orientation preferences that were rotated by 90 deg from radial (*Figure 4*, bottom row). Furthermore, on a

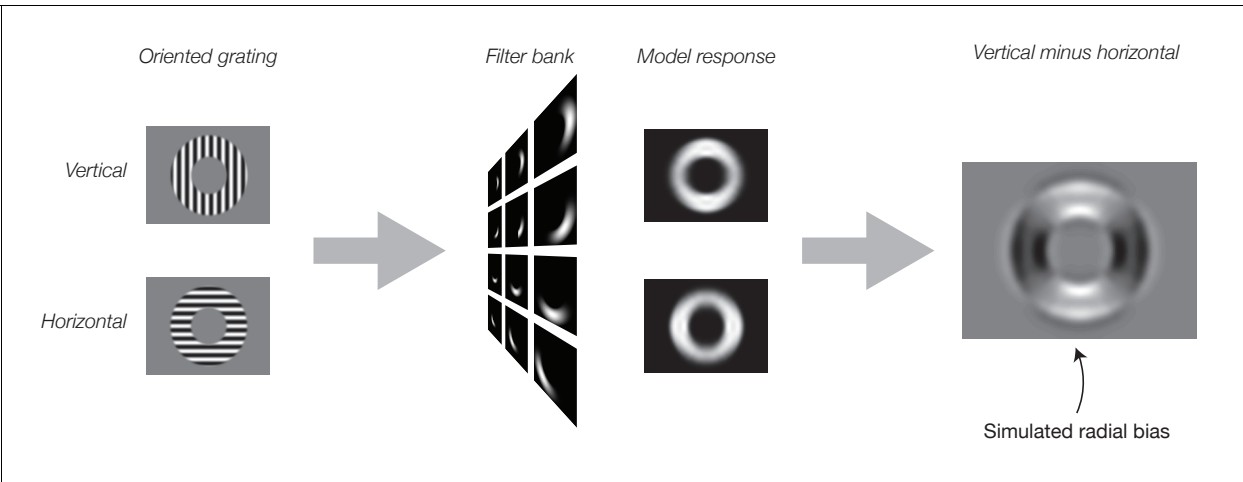

**Figure 2.** Simulated neural responses illustrating the impact of stimulus vignetting. Oriented gratings were vignetted by inner and outer annuli, using stimulus parameters identical to those from *Freeman et al. (2011)*. Next, the gratings were used as input to the model. Model responses were computed separately for vertical (top) and horizontal (bottom) gratings. Finally, model output was computed as the responses to vertical minus responses to horizontal gratings. Model exhibits a preference for horizontal gratings along the horizontal meridian and a preference for vertical gratings along the vertical meridian (i.e. a radial bias).

DOI: https://doi.org/10.7554/eLife.37241.003

voxel-by-voxel basis, preferred orientations were shifted by 90 deg across modulators (*Figure 5*). Comparing model predictions to data from individual voxels, we observed a clear correspondence between predicted and measured orientation preference (*Figure 6*). Circular correlations between predicted and measured orientation preferences were 0.26 and 0.20 for radial and angular

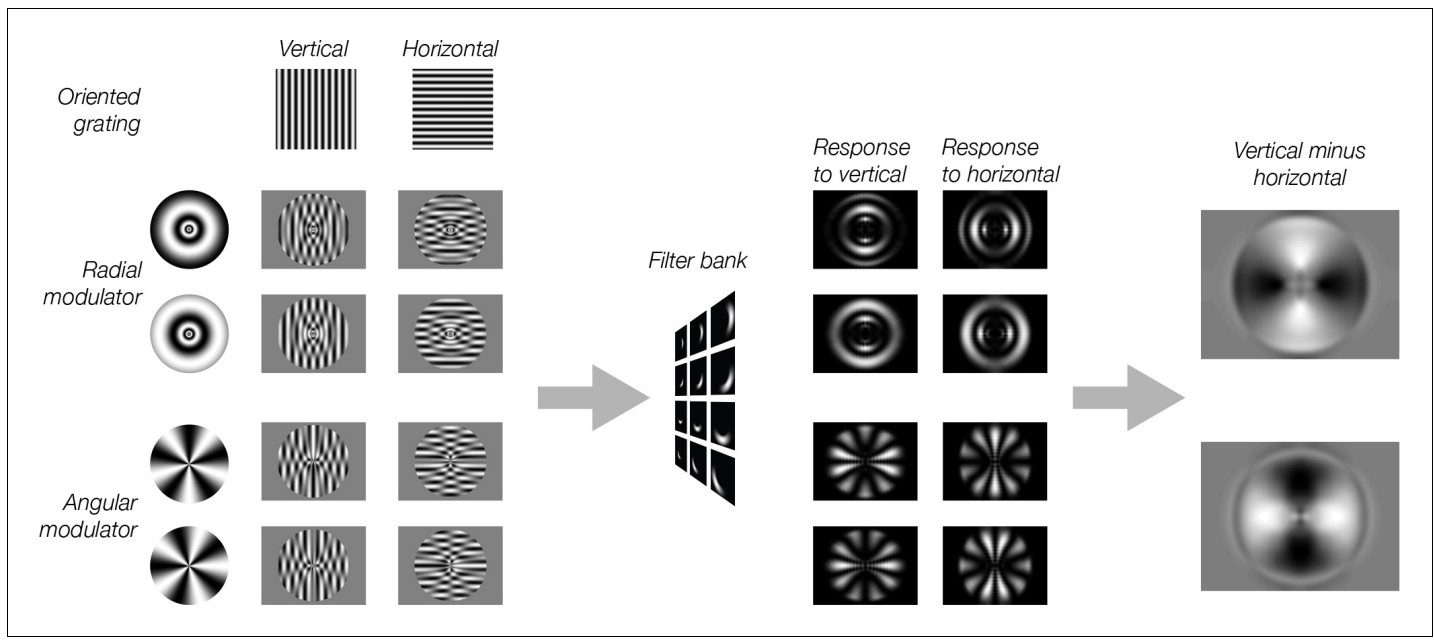

**Figure 3.** Model predictions for modulated gratings. Each of the eight stimuli were created by multiplying a vertical or horizontal grating by a radial or angular modulator. These stimuli were used as input to the model. Model responses were computed as in *Figure 2*. For radial modulated gratings (top two rows), the model exhibits a radial preference: larger responses to horizontal gratings along the horizontal meridian, larger responses to vertical gratings along the vertical meridian. However, for angular modulated gratings (bottom two rows), the orientation preference is tangential: larger responses for horizontal gratings along the vertical meridian and larger responses for vertical gratings along the horizontal meridian.

DOI: https://doi.org/10.7554/eLife.37241.004

modulators, respectively (p<0.0001 for both). Note that many sources of noise in the measurements could impact this analysis: in estimating the location and size of each voxel's pRF, in measuring each voxel's orientation preference, and in coregistering each voxel's response across multiple fMRI sessions. Moreover, the model we use is highly simplified; for example, it does not take into account changes in spatial frequency tuning at greater eccentricities. Yet, despite the multiple sources of noise and the simplified assumptions of the model, the correspondence between the model's prediction and the empirical measurements is highly statistically significant.

We hypothesized that if stimulus vignetting accounts for the radial bias, and changing the direction of the modulator flips voxels' orientation preferences by 90 deg, then the population pattern of activity should also shift by 90 deg. We tested this hypothesis by training a classifier on fMRI responses to the radial modulator, and testing how accurately it decodes responses to the angular modulator, and vice versa. We compared between-modulator decoding accuracy to accuracy obtained from training and testing with data from the same modulator (within-modulator). Within-

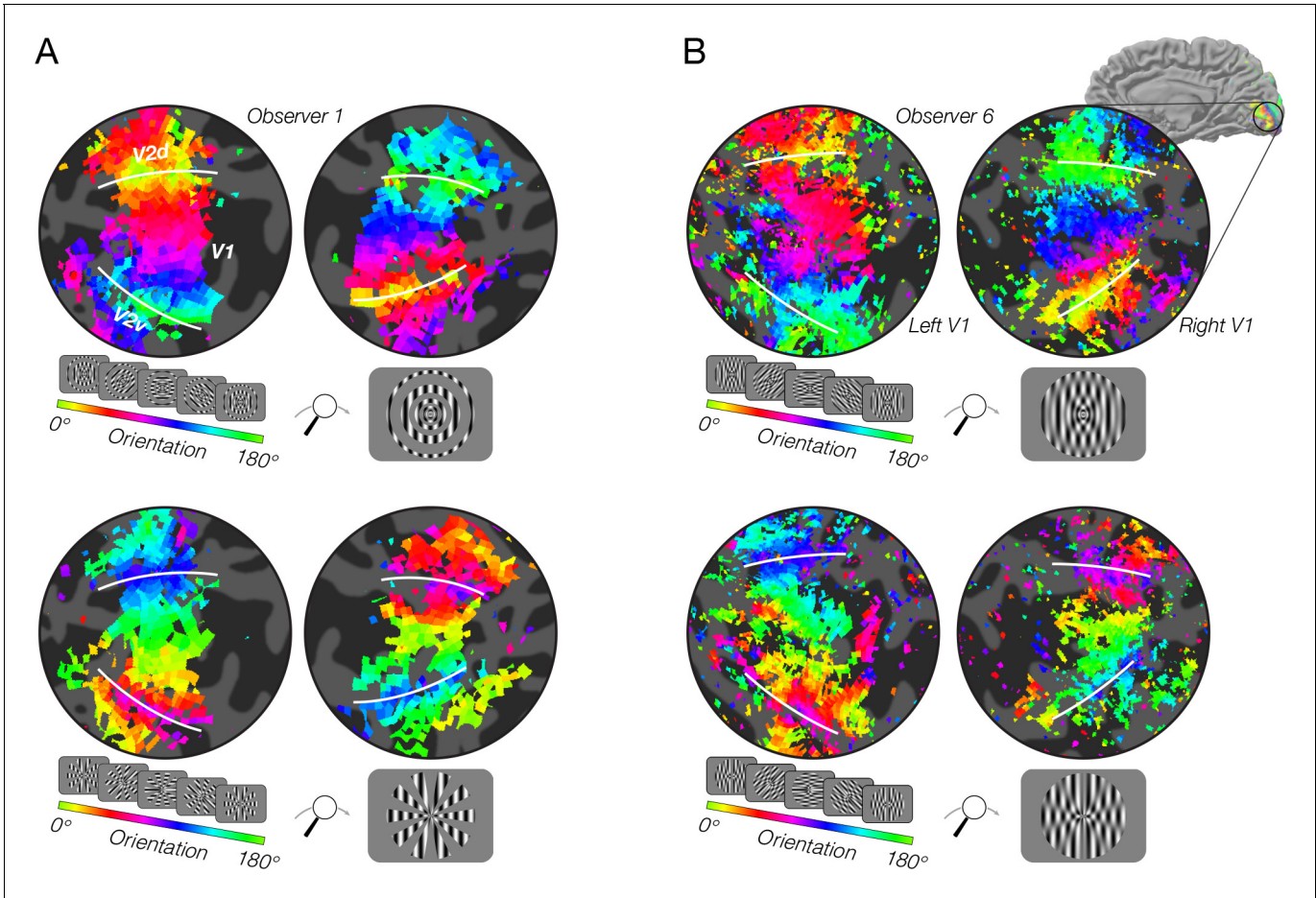

**Figure 4.** fMRI measurements of orientation bias depend on stimulus vignetting. (A) Conventional resolution, 3T. Top: Responses to phase-encoded oriented gratings multiplied by a static radial modulator (shown in inset). The oriented grating cycled through 16 steps of orientation ranging from 0° to 180° every 24 s. The radial modulator was constant during the entire experiment. Map thresholded at coherence of 0.2. Hue indicates phase of the best fitting sinusoid. White lines indicate V1/V2 boundaries. Bottom: Responses to the same oriented grating as in A, but here the grating is multiplied by an angular modulator. As predicted by the model, the radial modulator gave rise to a radial orientation bias, while the angular modulator gave rise to a tangential orientation bias. (B) High-resolution, high field strength (7T) measurements of orientation preference for radial and angular modulators. Stimuli and conventions same as for A, except the modulators were radial and angular sinusoids.

DOI: https://doi.org/10.7554/eLife.37241.005

The following figure supplement is available for figure 4:

**Figure supplement 1.** fMRI responses for all individual subjects.

DOI: https://doi.org/10.7554/eLife.37241.006

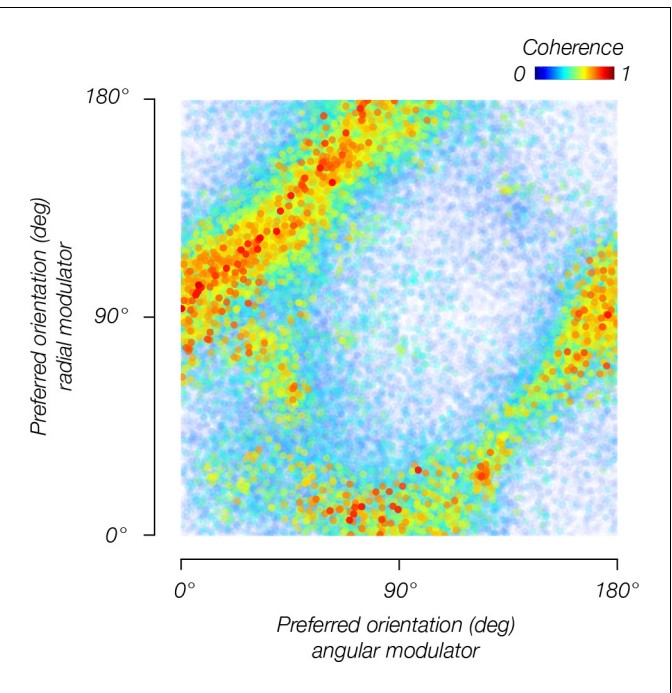

**Figure 5.** Orientation bias depends on the modulator. Orientation bias for V1 voxels pooled across observers for the angular modulator (y-axis) and radial modulator (x-axis).

DOI: https://doi.org/10.7554/eLife.37241.007

modulator decoding yielded high accuracies (*Figure 7A*, left, p<0.001, tested with a non-parametric permutation test). Across-modulator decoding accuracy was below chance (*Figure 7A*, middle, p<0.0001). These results indicate that the population response to oriented gratings was dependent on the modulator — responses to an oriented grating multiplied by one modulator (e.g. radial) were entirely different from responses to the same oriented grating when multiplied by the other modulator (e.g., by the angular modulator).

Does changing the modulator (between radial and angular) shift the population response pattern by 90 deg, as predicted by the model? To test this possibility, we repeated the between-modulator decoding analysis described above, but shifted the orientation labels by 90°. After shifting the orientation labels, decoding accuracy returned to levels significantly above chance (*Figure 7A*, right, p<0.0001), only slightly lower than within-modulator decoding accuracy on the unshifted data. The difference between within-modulator decoding accuracy and between-modulator decoding accuracy after shifting labels was small but statistically significant (p=0.046). Note that the two modulators share inner and outer radial edges. Thus, not all voxels are expected to shift by 90 deg. These common edges may be the cause of the slightly lower decoding accuracy across modulators. However, we cannot rule out the possibility that other sources of orientation-selective information not associated with the vignette, are present in the data.

We performed a complementary analysis, based on multidimensional scaling, to visualize the responses for both modulators in a single 2D plot (*Figure 7B*). Each line in *Figure 7B* represents a population response to a single stimulus orientation. The color and the line orientation in the plot both indicate the orientation of the stimulus. The distances between the lines reflect the dissimilarity between the corresponding population responses: similar responses are near to each other, and dissimilar responses are further apart. For each modulator, we found the responses to different orientations to be organized continuously in a ring. The two rings, corresponding to the two modulators, are misaligned with each other, reflecting the change in the population responses resulting from the change in modulator (*Figure 7B*). Specifically, responses to identical orientations (i.e. identical colored lines) are distant, while responses to dissimilar orientations are proximal. However, after rotating the labels for responses to the angular modulator by 90 deg, the two sets of responses are well

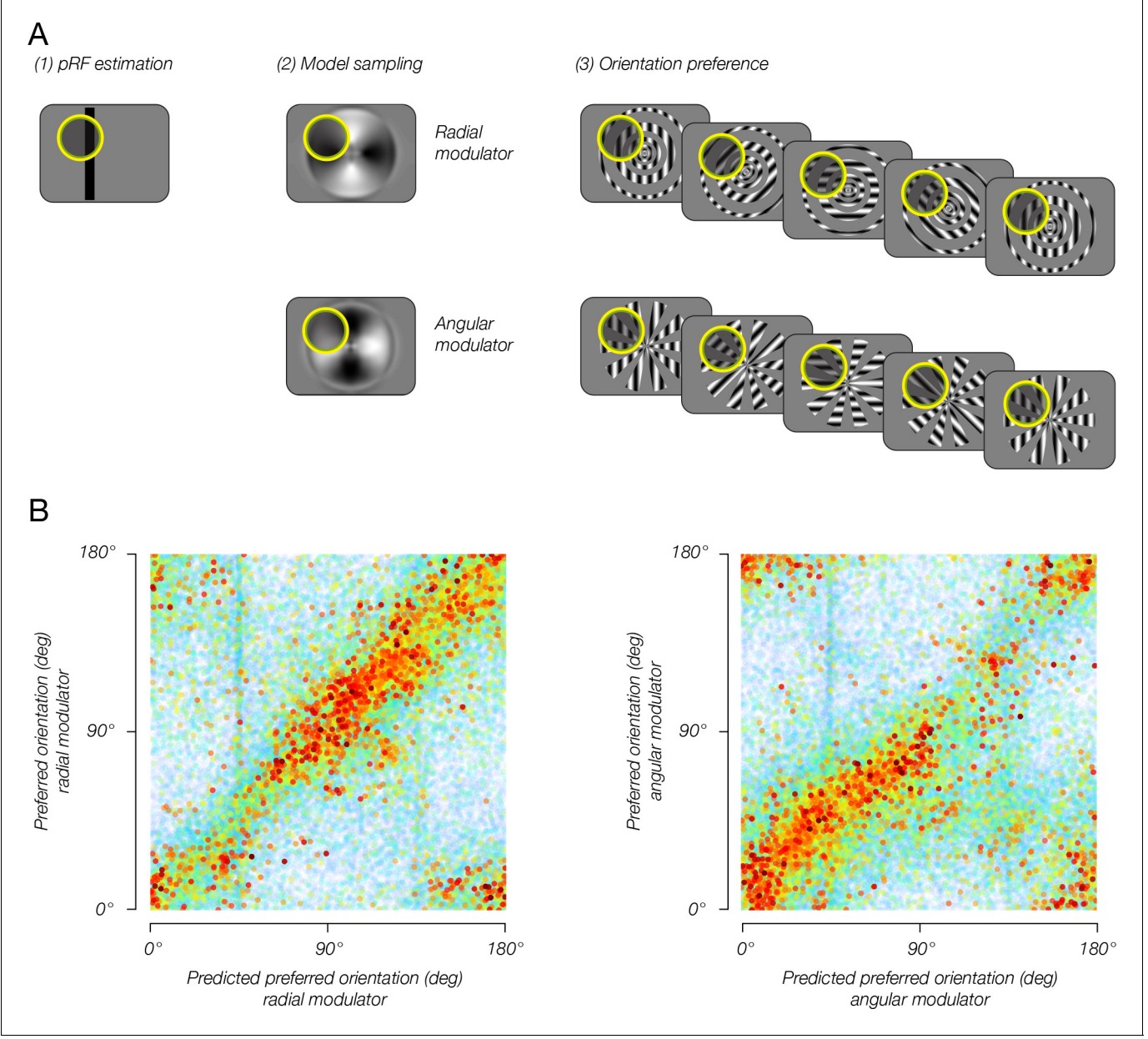

**Figure 6.** Combining the model with pRF estimates enables reliable predictions of single-voxel orientation preference. (**A**) pRF sampling method. (1) Each voxel's pRF was estimated based on an independent pRF-mapping scanning session. (2) Model output was sampled by the voxel pRF, resulting in an estimated preferred orientation for each modulator. (3) Estimated preferred orientation was compared to measured orientation preference, for each modulator. (**B**) Estimated orientation preference plotted against measured orientation preference. Sampling the model output with estimated pRFs yielded accurate predictions for each voxel's orientation preference for each of the two modulators, indicating that the model of stimulus vignetting provides a good account for the fMRI measurements of orientation preference.

DOI: https://doi.org/10.7554/eLife.37241.008

aligned. This demonstrates once again that changing modulators completely flips voxels' orientation bias, and this is reflected throughout the V1 retinotopic map.

## Discussion

Our results show that stimulus vignetting strongly affects voxel orientation bias. This suggests that results from many fMRI studies attempting to measure neural orientation tuning properties may

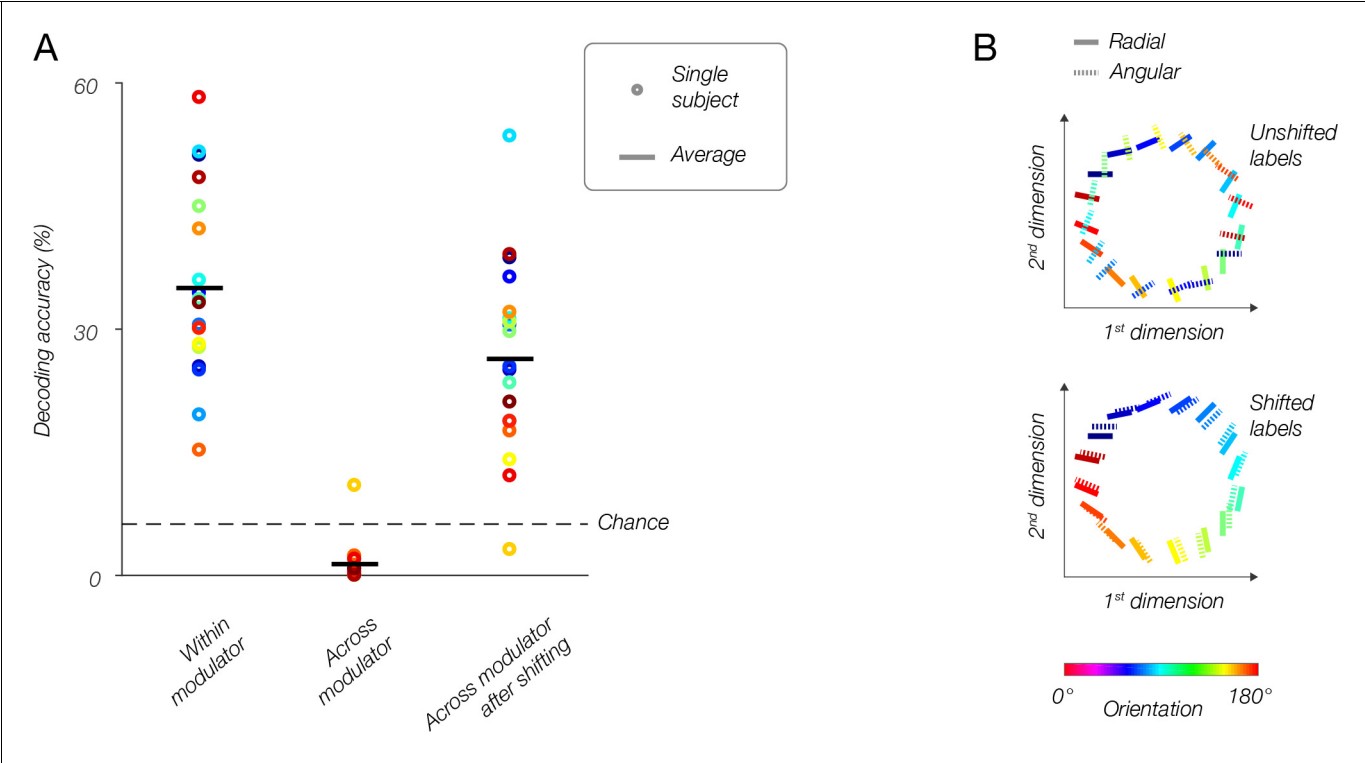

**Figure 7.** Population responses reflect the modulator, not stimulus orientation. (**A**) fMRI decoding. Within-modulator decoding: classifier was trained and tested on data from each modulator separately, in a leave-one-run-out cross-validation approach. Across modulator decoding: classifier was trained on data from one modulator and tested on data from the other modulator (and vice-versa). Across-modulator decoding after shifting: data for one modulator was shifted by 90°, then the classifier was trained on that modulator and tested on the other (and vice-versa). (**B**) Multidimensional scaling. Top: population responses to all orientations for both modulators (solid lines, radial modulator; dotted lines, angular modulator) are plotted in 2D, such that distances between conditions reflect how different (i.e., dissimilarity = 1–r, where r is the correlation coefficient) the responses are. For the original data, orthogonal orientations from the two modulators have similar responses and are plotted near each other, while identical orientations taken from the two modulators evoked very different responses, and lay distant from each other. Bottom: after shifting the labels for the angular modulator by 90 deg, the two datasets align well, and similar orientations correspond to similar population responses regardless of the modulator.
DOI: https://doi.org/10.7554/eLife.37241.009

actually reflect stimulus vignetting. Since stimulus vignetting masquerades as orientation selectivity amongst voxels that correspond to the stimulus edge, and because edges tend to be continuous over a large spatial extent, vignetting will generally result in a coarse-scale map of orientation bias. In many fMRI studies, gratings are presented within a circular annulus (*Haynes and Rees, 2005*; *Kamitani and Tong, 2005*; *Harrison and Tong, 2009*; *Freeman et al., 2011*; *Sun et al., 2013*; *Larsson et al., 2017*; *Wardle et al., 2017*), and in these cases the model predicts a radial coarse-scale bias. However, in some cases, other stimulus vignetting exists, resulting in more subtle coarse-scale patterns of orientation bias (*Swisher et al., 2010*; *Larsson et al., 2017*; *Sengupta et al., 2017*). One study (*Alink et al., 2017*) used inner and outer circular annuli, but added additional angular edges, the result of which should be a combination of radial and tangential biases. Indeed, this study reported that voxels had a mixed pattern of selectivity, with a considerable number of voxels reliably preferring tangential gratings, and other voxels reliably favoring radial orientations.

Stimulus vignetting is a general issue of concern in visual neuroscience (*Carter and Henning, 1971*; *Larsson et al., 2017*; *Tang et al., 2018*). According to the theoretical framework that we have applied here, whenever a neuron's receptive field overlaps a stimulus edge or a change in contrast, stimulus vignetting will spread the Fourier power and affect the neuron's response. As a result, such a neuron may exhibit ostensible orientation tuning even when it is not orientation selective. Furthermore, even if a neuron is truly orientation selective, measurements of its orientation tuning may be confounded by the aperture shape.

Vignetting may produce ostensible selectivity to second-order features, including vignettes themselves. For example, consider the modulated gratings that we used in the current study. A V1 neuron with a receptive field near the vertical meridian (i.e., near the V1/V2 border) might respond more to a radial modulated vertical grating than to an angular modulated vertical grating simply as a result of vignetting. This could be misinterpreted as sensitivity to second-order orientation contrast (*Zhou and Baker, 1993*; *Mareschal and Baker, 1998*; *Song and Baker, 2007*), when in fact the neuron is sensitive only to first-order orientation. There are no so-called 'second-order filters' in our model. Yet it exhibits orientation-selective biases for 2nd-order modulation because of vignetting. Similarly, vignetting might account for ostensible selectivity to orientation of illusory contours and second-order patterns, in both single-cell (*Grosof et al., 1993*) and fMRI studies (*Larsson et al., 2006*; *Montaser-Kouhsari et al., 2007*).

Vignetting may even produce ostensible orientation selectivity in neurons that are not orientation selective at all. A major question concerning the neural origin of orientation selectivity is whether and to what degree lateral geniculate nucleus (LGN) responses are tuned to orientation (*Ling et al., 2015*; *Sun et al., 2004*; *Sun et al., 2016*). Recently, one group addressed this question using fMRI. They found that orientation can be decoded from LGN, suggesting that thalamic responses are tuned to orientation (*Ling et al., 2015*), perhaps inheriting their selectivity from the retina (*Schall et al., 1986*). Another group, using calcium imaging to measure bouton responses in mice, found extensive orientation tuning as well (*Sun et al., 2016*). But stimulus vignetting may have played a critical role in both of these studies. In the former study (*Ling et al., 2015*), gratings were presented within a circular annulus, likely yielding a radial bias due to stimulus vignetting. In the latter study (*Sun et al., 2016*), the grating covered the entire visual display (75 deg x 75 deg), which is nearly the entire extent of the mouse monocular visual field. Hence, it might seem that vignetting would not have been a dominant source of orientation selectivity in this experiment. However, mouse neuronal receptive fields are large, at least an order of magnitude larger than in primates (*Van den Bergh et al., 2010*; *Tang et al., 2016*). A significant portion of neurons recorded by *Sun et al. (2016)* may have therefore had receptive fields that overlapped the edges of the screen, which would have resulted in vignetting. These and many other related studies may not have been measuring orientation selectivity at all, but rather activity reflecting stimulus vignetting.

Do voxel responses contain any orientation preferences that are not the result of stimulus vignetting? We found no evidence for any other orientation information present in the data. Based on the current study we cannot rule out the possibility of other factors affecting fMRI orientation preferences, at both coarse and fine spatial scales. For example, fine-scale signals originating from a columnar organization may, theoretically, affect fMRI signals (*Boynton, 2005*). But these columnar signals may be entirely eclipsed by stimulus vignetting. Similarly, coarse-scale signals such as a physiological radial bias may indeed exist, but be relatively small, and hence also eclipsed by stimulus vignetting. Our previous results, describing a vertical bias near the fovea (*Freeman et al., 2013*), in addition to the radial bias in the periphery (*Freeman et al., 2011*), lend support to the possibility that there are multiple sources of coarse-scale bias.

Do fMRI voxels reflect any fine-scale orientation signals originating from a columnar organization, in addition to an orientation bias resulting from vignetting? If each voxel reflects the activity of a large number of neurons, signals from multiple orientation columns will be summed within a voxel, thus attenuating any columnar signal. If this is the case, decreasing the voxel size should lower the number of different columns summed within each voxel, and amplify columnar orientation selectivity. On the other hand, orientation bias resulting from vignetting should not depend on scanning resolution. Nearby neurons respond to nearby locations in the visual field, and therefore will respond similarly to the vignette. Thus, increasing the acquisition resolution will augment columnar signals but not vignetting signals. Our results at 2 mm voxels demonstrate primarily vignetting with no evidence of a columnar signal. Increasing the acquisition resolution to 0.9 mm did not alter this result. From this we conclude that the coarse-scale bias due to vignetting is the predominant source of orientation information measured with fMRI at scanning resolutions with voxel sizes of 1 mm or larger.

We are no longer arguing about whether orientation decoding is dominated by coarse-scale orientation biases; we already know that it is. An extreme method we previously used for testing whether fine-scale information contributes to fMRI orientation decoding involved shifting the acquired slices by half a voxel (1 mm), and then attempting to decode the grating orientation (*Freeman et al., 2013*). If successful decoding depends on fine-scale information, shifting the slices

should eradicate any decoding information. In fact, decoding accuracy was unaffected (statistically indistinguishable) by shifting the slices. However, the finding that a coarse-scale bias is the source of orientation decoding remains controversial, and several recent studies have attempted to disprove it (*Alink et al., 2013*; *Pratte et al., 2016*; *Alink et al., 2017*), in part, we believe, because the notion that fMRI is sensitive to fine-scale neural activity is highly attractive. We have no doubt that there are orientation columns in human V1. But we also have no doubt that orientation-decoding with fMRI at conventional resolutions is dominated by coarse-scale orientation bias, not the fine-scale (columnar) architecture for orientation selectivity. Otherwise, decoding would have suffered from shifting the slices by half the hypercolumn width. Rather than continuing to debate about whether fMRI decoding is dominated by coarse- or fine-scale information, the current study characterizes one of the sources of the coarse-scale bias, the implications of which are much broader than the technical details of orientation-decoding with fMRI (see above).

Leaving aside this debate about whether orientation decoding with fMRI depends on coarse- vs. fine-scale information, it is important to remember that the contribution of many decoding studies remains, irrespective of the source of decodable information. For example, a primary contribution of *Kamitani and Tong (2005)* regards feature-based attention: they showed that attending to one or the other orientation changes the activity in V1 in such a way that the attended orientation can be decoded. That result holds regardless of whether the decoding depends on coarse- vs. fine-scale. Similarly, *Haynes and Rees (2005)* showed that V1 activity reflects properties of invisible stimuli, using decoding. Likewise for many studies that used decoding, forward modeling, and other MVPA techniques to reveal properties of cortical processing. Most, if not all, of these findings do not depend on the source of decodable information.

Stimulus vignetting has previously been referred to as an 'edge effect', and we demonstrated the intuition of stimulus vignetting using examples of hard edges (*Figure 1*). Several studies have used gratings with smoothed edges, and some have smoothed the edges in an attempt to eliminate any effects related to the edge (*Freeman et al., 2011*; *Warren et al., 2014*). According to the model, however, vignetting does not occur exclusively in the vicinity of edges. Rather, a contrast change, even a gradual change, is all that is necessary to produce stimulus vignetting. The vignette causes the Fourier spectrum to splatter (*Figure 1*). Smoothing the stimulus edge only acts to spatially spread this effect across a larger portion of the visual field. Indeed, we found that both square-wave modulators and sinusoidal modulators evoked similar levels of vignetting, empirically confirming the model's prediction.

Other factors, in addition to stimulus vignetting, may contribute to the coarse-scale bias. First, the coarse-scale orientation bias may reflect, in addition to stimulus vignetting, a neuronal 'gain map' (*Philips and Chakravarthy, 2016*). Neurons in visual cortex may have different response gains depending on their preferred orientations and receptive field locations. For example, the responses of a population of neurons tuned for vertical might be highest for those neurons with receptive fields near the vertical meridian and smallest for those along the horizontal meridian, and vice versa for neurons tuned for horizontal, conferring a cardinal bias. A gain map could be instantiated in visual cortex through a number of different physiological mechanisms. For example, in a population of neurons along the horizontal meridian, a gain map might be instantiated by (1) a larger proportion of neurons that prefer horizontal orientations, or (2) higher firing rates for the sub-population of neurons that prefer horizontal orientations, or (3) a wider tuning bandwidth for the sub-population of neurons that prefer orientations close to vertical. Each of these three mechanisms would give rise to a larger population response for horizontal gratings in fMRI voxels that lay along the horizontal meridian.

Second, the coarse-scale orientation bias might reflect 'asymmetric surround suppression', an imbalance in suppression between different parts of the region surrounding a neuron's receptive field. The responsiveness of a V1 neuron is suppressed by stimuli outside its classical RF (*Sceniak et al., 1999*; *Sceniak et al., 2001*; *Cavanaugh et al., 2002a*; *Cavanaugh et al., 2002b*; *Zenger-Landolt and Heeger, 2003*). Suppression is stronger for neurons with RFs near the center of a stimulus aperture, less so for neurons with RFs near the aperture edge, because there is less stimulation surrounding those RFs. For many neurons in V1, surround suppression is asymmetric, such that there is more suppression at the 'ends' of the receptive field (i.e., along the preferred orientation) than the 'sides' (*Larsson et al., 2017*; *Cavanaugh et al., 2002a*; *Cavanaugh et al., 2002b*; *Tanaka and Ohzawa, 2009*; *Hallum and Movshon, 2014*). Because of this asymmetry, the release

from suppression may be greater for aperture edges that are orthogonal to each neuron's preferred orientation, such that a circular aperture centered at fixation will confer a radial bias.

Regardless of whether a gain map and/or asymmetric surround suppression contribute to coarse-scale orientation bias, stimulus vignetting shows a strong and robust effect on measured orientation preference. Orientation is used as a model for understanding more complex feature processing in other cortical areas, and oriented V1-like receptive fields play an important role in successful computational models of vision. Yet, the computations that give rise to orientation tuning in V1 remain inadequately understood. Our results provide a framework for reinterpreting a wide-range of findings on orientation selectivity.

## Materials and methods

### Theoretical model

The image-computable model is based on the steerable pyramid (*Simoncelli et al., 1992*), a sub-band image transform that decomposes an image into orientation and spatial frequency channels (*Figure 1*). The pyramid simulates the responses of a large number of linear receptive fields (RFs), each of which computes a weighted sum of the stimulus image; the weights determine the orientation and spatial frequency tuning. RFs with the same orientation and spatial-frequency tuning, but shifted to different locations, are called a 'channel'. In the model, the number of spatial frequency channels, orientation channels, and orientation bandwidth are model parameters that can each be chosen flexibly. We used an instantiation of the model with six orientations, which resulted in a bandwidth comparable to empirical tuning curves measured in primate electrophysiological recordings (*Ringach et al., 2002*). Using four or more bands with correspondingly broader or narrower tuning curves yielded similar results supporting the same conclusions. The number of spatial frequency channels was determined by the size of the input images and the spatial frequency bandwidth. We used images that were 768 × 1024 pixels, and a spatial frequency bandwidth of 0.5 octaves, and so the model had 15 different spatial frequency channels. The RFs cover all orientations and spatial frequencies evenly (i.e., the sum of the squares of the tuning curves is exactly equal to one for all orientations and spatial frequencies). For each orientation and spatial frequency, the pyramid includes RFs with two different phases, analogous to odd- and even-phase Gabor filters. The sum of the squares of the responses of two such RFs computes what has been called an 'energy' response (*Adelson and Bergen, 1985*; *Heeger, 1992*) because it depends on the local spectral energy within a spatial region of the stimulus, for a particular orientation and spatial frequency. The energy responses are also arranged in channels. The linear RFs in the model are hypothesized to be a basis set of orientation and spatial frequency tuning curves of V1 neurons; any invertible linear transform of the basis set can be substituted (*Freeman and Adelson, 1991*; *Simoncelli et al., 1992*; *Simoncelli, 1993*) and different such transforms can be substituted at each spatial location, consistent with the diversity in the tuning properties of V1 neurons, without changing the nature of the representation. The model was designed so that the filters evenly tile retinotopic space, orientation, and spatial frequency to avoid any possibility of the kind of artifact that was suggested to underlie *Carlson (2014)* simulation results (*Clifford and Mannion, 2015*).

The strength of the model is its simplicity. Our goal is not to provide an accurate model of V1 activity, but rather to provide a platform for assessing vignetting. We purposely kept the model simple so that we could be certain that vignetting is indeed the source of orientation bias in the model. Ostensibly interesting features of V1 responses (e.g. second-order contrast sensitivity) might actually be a result of vignetting. If we were to include such features in the model, any orientation bias could be a combined result of vignetting and those features. Keeping the model simple eliminates that possibility.

We used the theoretical model to predict fMRI responses to a wide range of stimuli. We then chose stimuli that were predicted to have opposite patterns of orientation bias when measured with fMRI. Predicted responses were calculated by finding the scale with maximal responses, summing the energy responses for the channels at that scale, and averaging across stimulus grating phases (See: fMRI experiments, Stimuli).

## fMRI experiments

fMRI experiments were conducted at two sites. Experiments at 3T were conducted at NYU Center for Brain Imaging. Experiments at 7T were conducted at Functional Magnetic Resonance Imaging Core Facility at NIH. Methods were similar between the two sites except where noted.

### Stimuli

Stimuli consisted of two gratings (a carrier and a modulator) multiplied by one another (*Figure 3*). The carrier grating consisted of a large, oriented sinusoidal Cartesian grating (spatial frequency, NYU: 0.5 cycles per degree; NIH: 1.4 cycles per degree) presented within an annulus (NYU: inner radius, 0.5°; outer radius 9.5°; NIH: inner radius 0.75°; outer radius 9°). The spatial phase of the carrier grating was randomized every 250 ms from a predefined set of 16 phases uniformly distributed between 0 and $2\pi$. The orientation of the carrier grating cycled through 16 evenly- spaced angles between 0° and 180° (1.5 s per orientation). The carrier grating completed 10.5 cycles in each run. Each cycle was 24 s long, producing a run length of 4 min, 12 s. The orientation of the carrier grating cycled clockwise in half of the runs and counter-clockwise in the other half.

The modulator grating was a polar-transformed grating. In the 'square wave modulator' experiments (at NYU and NIH), the modulator grating was square, with hard edges. In the 'sinusoidal modulator' grating experiment (NIH), the modulator did not have any edges. On half the runs the modulator had a radial orientation, and an angular orientation in the other half. The modulator grating produced either a set of rings emanating from the fovea (radial orientation) or a set of inward-pointing wedges encircling the fovea (angular orientation). In the NIH experiments, the rings were scaled with eccentricity. On half the runs, the modulator grating was sine phase and cosine phase in the other half. We tested every combination of clockwise/counter-clockwise cycling carrier gratings, radial/angular modulator gratings, and sine/cosine phase modulator gratings. This produced a total of 8 conditions. Each of the eight conditions was tested in two independent runs, producing a total of 16 runs per scanning session.

Because the only stimulus feature that varied within each run was the orientation of the carrier grating, any fMRI activity measured within each run could be attributed to either the orientation of the carrier grating, or an interaction between the orientation of the carrier grating and the static modulator grating. The modulator itself did not evoke any modulation in the fMRI responses because it did not change during the course of a scanning run.

### Observers

Thirteen observers (eight females, aged 22–27 years) with normal or corrected-to-normal vision participated in the study. Observers provided written informed consent.

The consent and experimental protocol were in compliance with the safety guidelines for MRI research, and were approved by both the University Committee on Activities involving Human Subjects at New York University, and the Institutional Review Board at National Institutes of Health.

Each observer participated in multiple scanning sessions. Five observers (O1–O5) each participated in one session of the 'square wave modulator' experiment on the 3T scanner at NYU. Seven observers (O6–O11, O13) each participated in two sessions at NIH: one session for the 'square wave modulator' experiment and one session for the 'sine wave modulator' experiment. Data from a single session were discarded due to severe ghosting in the BOLD images. One observer (O12) did not participate in the 'sine wave modulator' experiment. In addition, each of the thirteen observers participated in one session to obtain a set of high-resolution anatomical volumes, used for cortical segmentation, and an additional session for retinotopic mapping.

### Orientation mapping experiments

Stimuli were generated using Matlab (MathWorks, MA) and MGL (*Gardner et al., 2018b*) on a Macintosh computer. Stimuli were displayed via an LCD projector (3T scanner: Eiki LC-XG250; resolution: 1024 × 768 pixels; refresh rate: 60 Hz. 7T scanner: PLUS U2-1200; resolution: 800 × 600 pixels; refresh rate: 60 Hz) onto a back-projection screen in the bore of the magnet. Observers viewed the display through an angled mirror at a viewing distance of approximately 58 cm (field of view in the 3T at NYU: 31.6 deg × 23.7 deg; field of view in the 7T at NIH: 20.5 deg × 16.1 deg).

The novel stimuli consisted of two gratings (a carrier and a modulator) multiplied by one another (*Figure 3*). The carrier grating consisted of a large, oriented sinusoidal Cartesian grating presented within an annulus. The spatial phase of the carrier grating was randomized every 250 ms from a pre-defined set of 16 phases uniformly distributed between 0 and $2\pi$. The orientation of the carrier grating cycled through 16 evenly-spaced angles between 0° and 180° (1.5 s per orientation). The carrier grating completed 10.5 cycles in each run. Each cycle was 24 s long, producing a run length of 4 min, 12 s. The orientation of the carrier grating cycled clockwise in half of the runs and counterclockwise in the other half.

## Behavioral task

Throughout each run, observers continuously performed a demanding two-interval, forced-choice task to maintain a consistent behavioral state and stable fixation, and to divert attention from the main experimental stimuli. In each trial of the task, the fixation cross (a 0.4 deg crosshair) dimmed twice (for 400 ms at a time), and the observer indicated with a button press the interval (1 or 2) in which it was darker. The observer had 1 s to respond and received feedback through a change in the color of the fixation cross (correct green, incorrect red). Each contrast decrement and response period was preceded by a 800 ms interval, such that each trial lasted 4.2 s. The fixation task was out of phase with the main experimental stimulus presentation. Contrast decrements were presented using an adaptive, 1-up-2-down staircase procedure (*Levitt, 1971*) to maintain performance at approximately 70% correct.

MRI data were acquired on two different scanners at two different sites. At NYU, data were acquired for observers O1-O5 on Siemens (Erlangen, Germany) 3T Allegra head-only scanner using a transmit head coil (NM-011, Nova Medical, Wakefield, MA), and an eight-channel phased-array surface receive coil (NMSC-071, Nova Medical). Functional imaging was conducted with 24 slices oriented perpendicular to the calcarine sulcus and positioned with the most posterior slice at the occipital pole (TR: 1500 ms; TE: 30 ms; FA: 72°; voxel size: $2 \times 2 \times 2.5$ mm; grid size: $104 \times 80$ voxels).

At NIH, data were acquired from observers O6-O13 on a research-dedicated Siemens 7T Magnetom scanner using a 32-channel head coil, located in the Clinical Research Center on the National Institutes of Health campus (Bethesda, MD). Functional imaging was conducted with 54 slices oriented perpendicular to the calcarine sulcus covering the posterior half of the brain (TR: 1500 ms; TE: 23 ms; FA: 55°; voxel size: $1.2 \times 1.2 \times 1.2$ mm or $0.9 \times 0.9 \times 0.9$ mm with 0% or 10% gap between slices, respectively; grid size: $160 \times 160$ voxels. Multiband factor 2 or 3, GRAPPA/iPAT factor 3).

For each observer, a high-resolution anatomy of the entire brain was acquired by co-registering and averaging between 2 and 8 T1-weighted anatomical volumes (magnetization-prepared rapid-acquisition gradient echo, or MPRAGE; TR: 2500 ms; TE: 3.93 ms; FA: 8°; voxel size: $1 \times 1 \times 1$ mm; grid size: $256 \times 256$ voxels). The averaged anatomical volume was used for co-registration across scanning sessions and for gray-matter segmentation and cortical flattening. Functional scans were acquired using T2*-weighted, gradient recalled echo-planar imaging to measure blood oxygen level-dependent (BOLD) changes in image intensity (*Ogawa et al., 1990*). At NYU, an MPRAGE anatomical volume with the same slice prescription as the functional images ('inplane') was also acquired at the beginning of each scanning session (TR: 1530 ms; TE: 3.8 ms; FA: 8°; voxel size: $1 \times 1 \times 2$ mm with 0.5 mm gap between slices; grid size: $256 \times 160$ voxels). The inplane anatomical was aligned to the high-resolution anatomical volume using a robust image registration algorithm (*Nestares and Heeger, 2000*).

At NIH, prior to the first experimental functional run of each session, 30 volumes were acquired with identical scanning parameters and slice prescription as the subsequent functional runs, except for the phase encoding direction which was reversed. This scan was used for correcting spatial distortions (see next section).

## fMRI time series preprocessing

The single reverse phase-encoded run was used to estimate the susceptibility-induced off-resonance field using a method similar to that described in (*Andersson et al., 2003*) as implemented in FSL (*Smith et al., 2004*). This estimate was then used to correct the spatial distortions in each subsequent run in the session. Data from the first half cycle (eight volumes) at the beginning of each functional run were discarded to minimize the effect of transient magnetic saturation and to allow

hemodynamic response to reach steady state: the first half cycle. Functional data were compensated for head movement within and across runs (*Nestares and Heeger, 2000*), linearly detrended, and high-pass filtered (cutoff: 0.01 Hz) to remove low-frequency noise and drift (*Smith et al., 1999*). The time series for each voxel was divided by its mean to convert from arbitrary intensity units to percent change in image intensity. Time series data from each run were shifted back in time by three volumes (4.5 s) to compensate approximately for hemodynamic lag. Time series for the clockwise runs were time reversed to match the sequence of the counterclockwise runs. Time series were then averaged within modulator type (the matrix of 8 conditions: cosine and sine modulator crossed with clockwise and counterclockwise orientation). Each individual voxel's time-course was fitted to a cosine, and the orientation corresponding to the phase of the cosine was assigned as that voxel's preferred orientation. It should be noted that shifting, inverting and averaging minimizes differences in hemodynamic responses between voxels, and results in a smooth time series that fits well to a cosine. Without this stage, the time series would have less Fourier power at the fundamental frequency and much of the power would instead shift to harmonic frequencies (see (*Pratte et al., 2016*) for examples of these effects).

## Retinotopic maps

Retinotopy was measured using non-periodic traveling bar stimuli and analyzed using the population receptive field (pRF) method (*Dumoulin and Wandell, 2008*). Bars were 3 deg wide and traversed the field of view in sweeps lasting 24 s. Eight different bar configurations (4 orientations and two traversal directions) were presented. The pRF of each voxel was estimated using standard fitting procedures (*Dumoulin and Wandell, 2008*), implemented in Matlab using mrTools (*Gardner et al., 2018a*). Visual area boundaries were drawn by hand on the flat maps, following published conventions (*Engel et al., 1997*; *Larsson et al., 2017*; *Wandell et al., 2007*).

## pRF sampling analysis

To test the model's predictions for the responses of individual fMRI voxels, we estimated pRFs and sampled the model's output at the corresponding region of the visual field. Specifically, each voxel's Gaussian pRF was used to weight the model output for each grating orientation, averaged across grating phase and modulator phase, resulting in a single predicted response value for each orientation. We then fitted a cosine to the responses, with the phase reflecting the voxel's predicted preferred orientation. This was done separately for each of the two modulators. Finally, we compared the predicted preferred orientation to the measured preferred orientation (see *Figure 6*, top).

## fMRI MVPA analysis

Classification was performed on fMRI time series, in a leave-one-run-out manner, with a naive Bayes classifier (using Matlab function 'classify' with 'diagLinear' option). First, for each voxel, we averaged across all 10 cycles within a run, yielding 16 time points per voxel, which reflected the amplitude of the response to each stimulus orientation. We refer to these 16 vectors across voxels as population responses, a single response vector per time point, or equivalently, per orientation.

For within-modulator decoding, the classifier was trained on data (response vectors and corresponding orientation labels) from seven runs, and tested on the 8th run (leave-one-run-out decoding). If there were consistent differences in the responses to different orientations the classifier should decode correctly at an accuracy rate that is above chance level. We then cross-validated, by repeating this procedure eight times, each time leaving out a different run for testing. This was repeated for both modulators, and results were averaged across them.

For across-modulator decoding, the classifier was trained on data from one modulator, then tested on data from the other modulator. We then swapped the training and testing datasets and repeated the analysis. For shifted across-modulator decoding, we first shifted the orientation labels for the angular modulator by 90 deg, then repeated the across-modulator decoding procedure, and averaged the accuracy. Statistical significance of decoding accuracy was determined by randomly permuting the phases of the Fourier transform of the raw time-course 10,000 times and comparing the mean accuracy across subjects to the mean accuracy without permuting. Decoding accuracy $X$ was deemed significant at $p<a$ if the fraction of permutations that yielded decoding accuracies higher than $X$ was smaller than $a$.

While decoding enabled us to measure orientation information present in population responses, it provided only an indirect view of the shift the responses undergo, as a result of changing the modulator. To visualize the responses in a more direct fashion, we used representational similarity analysis. Each condition (i.e., 16 orientations for each of 2 modulators), represented by a population response vector averaged across cycles and runs, was correlated with all other conditions, yielding a correlation matrix. The matrix was transformed into a dissimilarity matrix by subtracting from 1 (i.e., dissimilarity = 1-$r$, where $r$ is the correlation coefficient). We then averaged across subjects and used multidimensional scaling (Matlab function 'mdscale') to reduce the dimensionality to 2. Results are plotted both with the original orientation labels, and after shifting the angular modulator's orientation labels by 90 deg.

## Data and code sharing

We make publicly available all of the fMRI data reported here, as well as open-source software for implementing the theoretic model of V1, generating stimuli for the fMRI experiments, and analyzing the fMRI data (*Merriam and Roth, 2018*), at https://doi.org/10.17605/OSF.IO/TBJRF.

## Acknowledgements

Supported by NIH grants R01-EY025673, and by the Intramural Research Program of the National Institutes of Health (ZIA-MH-002909) - National Institute of Mental Health Clinical Study Protocol 93-M-0170, NCT00001360. ZNR was supported by The ELSC Postdoctoral Fellowship Abroad, Hebrew University of Jerusalem. Special thanks to Eero Simoncelli, Chris Baker, Peter Bandettini for helpful comments.

## Additional information

### Funding

| Funder | Grant reference number | Author |
|---|---|---|
| National Institute of Mental Health | ZIA-MH-002909 | Zvi N. Roth<br>Elisha P. Merriam |
| Hebrew University of Jerusalem | ELSC Postdoctoral Fellowship Abroad | Zvi N. Roth |
| National Institutes of Health | R01-EY025673 | David J. Heeger<br>Elisha P. Merriam |

The funders had no role in study design, data collection and interpretation, or the decision to submit the work for publication.

### Author contributions

Zvi N Roth, Data curation, Formal analysis, Investigation, Visualization, Methodology, Writing–original draft, Writing–review and editing, Software, Validation; David J Heeger, Conceptualization, Software, Investigation, Writing—original draft, Writing—review and editing, Funding acquisition, Formal analysis; Elisha P Merriam, Conceptualization, Methodology, Software, Investigation, Resources, Writing—original draft, Writing—review and editing, Supervision, Project administration, Funding acquisition, Formal analysis, Data curation, Validation

### Author ORCIDs

Zvi N Roth (iD) http://orcid.org/0000-0002-2173-1625
David J Heeger (iD) https://orcid.org/0000-0002-3282-9898
Elisha P Merriam (iD) http://orcid.org/0000-0003-2787-566X

### Ethics

Human subjects: Experiments were conducted with the written consent of each subject and in accordance with the safety guidelines for fMRI research, as approved by the University Committee on Activities Involving Human Subjects at New York University and by the Institutional Review Board of the National Institutes of Health.

#### Decision letter and Author response
Decision letter https://doi.org/10.7554/eLife.37241.012
Author response https://doi.org/10.7554/eLife.37241.013

---

# Additional files

### Supplementary files
• Transparent reporting form
DOI: https://doi.org/10.7554/eLife.37241.010

### Data availability
We make publicly available all of the fMRI data reported here, as well as open-source software for implementing the theoretic model of V1, generating stimuli for the fMRI experiments, and analyzing the fMRI data (Merriam, 2018), at https://doi.org/10.17605/OSF.IO/TBJRF (copy archived at https://github.com/elifesciences-publications/stimulusVignetting).

---

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
