## [Decision Letter]

Thank you for submitting your article "Stimulus vignetting and orientation selectivity in human visual cortex" for consideration by *eLife*. Your article has been reviewed by three peer reviewers, including Floris P de Lange as the Reviewing Editor and Reviewer #1, and the evaluation has been overseen by David Van Essen as the Senior Editor. The following individuals involved in review of your submission have agreed to reveal their identity: Kendrick N Kay (Reviewer #2); Nikolaus Kriegeskorte (Reviewer #3).

The reviewers have discussed the reviews with one another and the Reviewing Editor has drafted this decision to help you prepare a revised submission.

Summary:

The manuscript uses computational modeling and experimental measurements to make a compelling case that stimulus edge effects (or 'vignetting') may be a substantial source of apparent orientation tuning as measured using standard-resolution fMRI. The work is technically solid and carefully presented, and the topic will be of significant interest to the fMRI community (given the widespread use of multivoxel pattern analysis).

Essential revisions:

I have provided a summary of essential revisions. You will find more details, as well as additional points that need to be addressed, in the original reviews (appended below).

1) Tone down claims of novelty:

-The paper claims to introduce a novel idea that requires reinterpretation of a large literature. The claim of novelty is unjustified. Vignetting was discovered by Carlson et al., 2014 and in Wardle et al., 2017, Carlson's group showed that it may be one, but not the only contributing factor enabling orientation decoding. Carlson et al. deserve clearer credit throughout. See reviewer #2 point 1, and reviewer #3.

2) Provide a more nuanced coverage of the literature. E.g., the conjecture that the orientation preferences in fMRI measurements arise primarily from random spatial irregularities in the fine-scale columnar architecture doesn't seem 'leading' anymore in 2018, and more nuanced positions have been articulated since. See reviewer #1 point 1 and reviewer #3.

3) Discuss the broader implications of the study: the authors claim that the study has wide implications for many studies that used decoding of oriented gratings. But it is left unspecified what those implications are. Could the authors be more specific? For example, how should we reinterpret Kamitani and Tong, 2005 or Haynes and Rees, 2005? What wrong conclusions have been drawn, if we accept the notion that stimulus vignetting is the source of orientation decoding?

The significance of the present work might be further emphasized by relating it more broadly to the general approach of MVPA (i.e. using linear classifiers to decode activity). I believe the larger lesson, highlighted by the present work, is that even seemingly simple properties are in fact hard to isolate experimentally and that powerful approaches like classification can pick up on aspects of the data that might not be what the experimenter intended. See reviewer #1 point 4, reviewer #2 point 2.

4) Address/discuss whether decoding is still possible in the absence of vignetting effects, or is solely dependent on vignetting. See reviewer #1 point 2, 3; reviewer #3.

Reviewer #1:

The authors test the hypothesis that the orientation selectivity of responses in visual cortex, measured at a macroscopic scale, is caused by 'vignetting', i.e. second order changes in luminance caused by the aperture within which an oriented grating is presented. Using a computational model of neural responses in V1, they demonstrate the impact of vignetting on the model response, and confirm predictions made by the model in human fMRI measurements.

I find this a compelling manuscript, with a clear question that is of interest to many researchers. The results are expertly analyzed and clear cut, showing that stimulus vignetting may indeed be a major contributor to the orientation selectivity measured with fMRI. I have some comments related to the coverage of the literature, and a query about the implications of the study for the field.

1) Introduction section: "A leading conjecture is that the orientation preferences in fMRI measurements arise primarily from random spatial irregularities in the fine-scale columnar architecture (Boynton, 2005; Haynes and Rees, 2005; Kamitani and Tong, 2005)."

This conjecture was perhaps leading in 2005, but I don't think this is an accurate description of the state of affairs 13 years later, and it therefore seems a bit of a straw man. For example, a more nuanced view articulated by Swisher et al., 2010 from the Tong lab states that orientation information "can be found at spatial scales ranging from the size of individual columns to about a centimeter". I suggest the authors paint a more balanced picture in the introduction.

2) Subsection “Coarse-scale bias and stimulus vignetting: fMRI experiments”: "From this, we conclude that stimulus vignetting is a primary source of the coarse scale bias."

Why “a primary source”, rather than “a source”, or 'an important source'? It seems bold to conclude this based on a correlation of ~0.2-0.26 between the model and the data?

3) In the same section: "Decoding accuracy for the shifted between-modulator analysis was only slightly lower than within-modulator decoding accuracy".

Could the authors test whether this difference is statistically significant?

If the difference is significant, this could, as the authors point out, be due to the inner and outer radial edges. However, it could potentially also be caused by the fact that there is a small amount of orientation information present in the fMRI activity patterns that is not due to stimulus vignetting. The authors may also want to include this possibility in the text – even though they may find it unlikely.

4) Discussion section: "Our results provide a framework for reinterpreting a wide-range of findings on orientation selectivity, measured with both fMRI in human subjects and in single units."

This statement suggests that the study has wide implications for many studies that used decoding of oriented gratings. But it is left unspecified what those implications are. Could the authors be more specific here? For example, how should we reinterpret Kamitani and Tong, 2005 or Haynes and Rees, 2005? What wrong conclusions have been drawn, if we accept the notion that stimulus vignetting is the source of orientation decoding?

Reviewer #2:

The manuscript by Roth et al. uses computational modeling and experimental measurements to make a compelling case that stimulus edge effects (or 'vignetting') may be a substantial source of apparent orientation tuning as measured using standard-resolution fMRI. The work is technically solid and carefully presented, and the topic will be of significant interest to the fMRI community (given the widespread use of multivoxel pattern analysis). I have some comments, mainly conceptual/framing in nature, which should be relatively easy to address.

1) Wording and framing.

- There are a few places where I think the conclusions and claims should be toned down. For example, "vast number of previous studies" (Introduction section) and "wide-range of findings in the visual system". I assume the authors are referring to past studies that have used orientation stimuli in fMRI, and not the neurophysiology literature on orientation tuning in single neurons. While it is theoretically possible that vignetting effects may be influencing single-neuron response properties (since Gabor and grating patches are widely used as stimuli), it is not yet clear whether single-neuron studies need to be re-interpreted.

- The work of Carlson, 2014 involves modeling work that is similar to what is done in the current manuscript. Of course, the major advance of the present work is the demonstration of empirical findings, but this previous work might deserve more acknowledgment.

- Ultimately, a lot of the controversy regarding fMRI orientation decoding comes down to numbers, and it would be helpful to clarify what is meant by "coarse" and "fine". I assume the authors mean something to the effect that "coarse" is > 1 mm and "fine" is < 0.5 mm (or something like that).

2) Some big-picture perspective.

- The significance of the present work might be further emphasized by relating it more broadly to the general approach of MVPA (i.e. using linear classifiers to decode activity). I believe the larger lesson, highlighted by the present work, is that even seemingly simple properties are in fact hard to isolate experimentally and that powerful approaches like classification can pick up on aspects of the data that might not be what the experimenter intended. One way that we have conceptualized this (Naselaris, TICS, 2015) is that the orientation of a grating stimulus is not the only stimulus feature that can give rise to variance in data, and that classification can reflect a number of different stimulus features, such as those related to vignette effects, unless one does work to rule them out, e.g. by considering explicit computational models.

3) Acknowledgment of the limitations of the model.

- I think the main contribution of the current paper is the experimental results. The modeling analyses do provide value in that they demonstrate a concrete (and reasonably plausible) explanation of what could be driving the observed orientation-tuning results. However, as the authors recognize in the Discussion, there are many stimulus properties beyond what is characterized in the model that are known to affect V1 responses (e.g. surround suppression, contour integration, 2nd-order contrast effects, etc.), and which might also contribute to the orientation effects. Thus, the text could be clarified to acknowledge the limitations of the model and indicate what role the modeling results play in this specific paper. It seems that the role of the modeling results is to show concretely that imbalances in filter energy across orientations exist at stimulus edges and that this is one possible reason for finding orientation tuning in standard-resolution fMRI. (Note that I am not suggesting that the present paper needs to perform detailed model comparisons (in which different models are pitted to quantitatively account for individual voxel responses to a variety of stimulus conditions); that would be outside of the scope of this paper.)

4) Clarification of the modeled effect.

- It would be helpful to isolate and clarify the nature of the effect shown in Figure 2. One potential explanation is that for a linear filter stimulated with an optimal oriented grating, the filter shows a bigger response when the grating has a vignette edge orthogonal to the orientation compared to when the grating has a vignette edge parallel to the orientation. Is this the case in the model?

Reviewer #3:

Vignetting: interactions between grating and aperture edges might explain coarse-scale orientation-preference maps in V1 [I6 R8].

The orientation of a visual grating can be decoded from fMRI response patterns in primary visual cortex (Kamitani and Tong, 2005, Haynes and Rees, 2005). This was surprising, because fMRI voxels in these studies are 2-3 mm wide in each dimension and thus average over many columns of neurons responding to different orientations. Since then, many studies have sought to clarify why fMRI orientation decoding works so well.

The first explanation given was that even though much of the contrast of the neuronal orientation signals might cancel out in the averaging within each voxel, any given voxel might retain a slight bias toward certain orientations if it didn't sample all the columns exactly equally (e.g. Boynto,n 2005, Kamitani and Tong, 2005). By integrating the evidence across many slightly biased voxels with a linear decoder, it should then be possible to guess, better than chance, the orientation of the stimulus.

Another account (Op de Beeck, 2010, Freeman et al., 2011) proposed that decoding may rely exclusively on coarse-scale spatial patterns of activity. In particular, Freeman Brouwer, Heeger and Merriam, 2011 argued that orientations that are radial (aligned with a line that passes through the fixation point) are over-represented in the neural population. If this were the case, then a grating would elicit a coarse-scale response pattern across its representation in V1, in which the neurons representing edges pointing (approximately) at fixation are more strongly active. There is indeed evidence from multiple studies for a nonuniform representation of orientations in V1 (Furmanski and Engel, 2000, Sasaki et al., 2006, Serences et al., 2009, Mannion et al., 2010), perhaps reflecting the nonuniform probability distribution of orientation in natural visual experience. The over-representation of radial orientations might help explain the decodability of gratings. However, opposite-sense spirals (whose orientations are balanced about the radial orientation) are also decodable (Mannion et al., 2009, Alink et al., 2013). This might be due to a simultaneous over-representation of vertical orientations (Freeman et al., 2013, but see Alink et al., 2013).

There's evidence in favor of a contribution to orientation decoding of both coarse-scale (Freeman et al., 2011, Freeman et al., 2013) and fine-scale components of the fMRI patterns (e.g. Shmuel et al., 2010, Alink et al., 2013, Pratte et al., 2016, Alink et al., 2017).

Note that both coarse-scale and fine-scale pattern accounts suggest that voxels have biases in favor of certain orientations. An entirely novel line of argument altogether was introduced to the debate by Carlson, 2014.

Carlson, 2014 argued, on the basis of simulation results, that even if every voxel sampled a set of filters uniformly representing all orientations (i.e. without any bias), the resulting fMRI patterns could still reflect the orientation of a grating confined to a circular annulus (as standardly used in the literature). The reason lies in "the interaction between the stimulus region and the empty background" (Carlson, 2014), an effect of the relative orientations of the grating and the edge of the aperture (the annulus within which the grating is visible). Carlson's simulations showed that the average response of a uniform set of Gabor orientation filter is larger where the aperture edge is orthogonal to the grating. He also showed that the effect does not depend on whether the aperture edge is hard or soft (fading contrast). Because the voxels in this account have no bias in favour of particular orientations, Carlson aptly named his account an "unbiased" perspective.

The aperture edge adds edge energy. The effect is strongest when the edge is orthogonal to the carrier grating orientation. We can understand this in terms of the Fourier spectrum. Whereas a sine grating has a concentrated representation in the 2D Fourier amplitude spectrum, the energy is more spread out when an aperture limits the extent of the grating, with the effect depending on the relative orientations of grating and edge.

For an intuition on how this kind of thing can happen, consider a particularly simple scenario, where a coarse rectangular grating is limited by a sharp aperture whose edge is orthogonal to the grating. V1 cells with small receptive fields will respond to the edge itself as well as to the grating. When edge and grating are orthogonal, the widest range of orientation-selective V1 cells is driven. However, the effect is present also for sinusoidal gratings and soft apertures, where contrast fades gradually, e.g. according to a raised half-cosine.

An elegant new study by Roth, Heeger, and Merriam, 2018 now follows up on the idea of Carlson, 2014 with fMRI at 3T and 7T. Roth et al. refer to the interaction between the edge and the content of the aperture as "vignetting" and used apertures composed of either multiple annuli or multiple radial rays. These finer-grained apertures spread the vignetting effect all throughout the stimulated portion of the visual field and so are well suited to demonstrate the effect on decodability.

Roth et al. performed simulations, following Carlson, 2014 and assuming that every voxel uniformly samples all orientations. They confirm Carlson's account and show that the grating stimuli the group used earlier in Freeman et al., 2011 is expected to produce the stronger response to radial parts of the grating, where the aperture edge is orthogonal to the grating. Freeman et al., 2011 used a relatively narrow annulus (inner edge: 4.5°, outer edge: 9.5° eccentricity from fixation), where no part of the grating is far from the edge. This causes the vignetting effect to create the appearance of a radial bias that is strongest at the edges but present even in the central part of the annular aperture. Roth et al.'s present findings suggest that the group's earlier result might reflect vignetting, rather than (or in addition to) a radial bias of the V1 neurons.

Roth et al. use simulations also to show that their new stimuli, in which the aperture consists of multiple annuli or multiple radial rays, predict coarse-scale patterns across V1. They then demonstrate in single subjects measured with fMRI at 3T and 7T that V1 responds with the globally modulated patterns predicted by the account of Carlson, 2014.

The study is beautifully designed and expertly executed. Results compellingly demonstrate that, as proposed by Carlson, 2014, vignetting can account for the coarse-scale biases reported in Freeman et al., 2011. The paper also contains a careful discussion that places the phenomenon in a broader context. Vignetting describes a family of effects related to aperture edges and their interaction with the contents of the aperture. The interaction could be as simple as the aperture edge adding edge energy of a different orientation and thus changing orientation selective response. It could also involve extra-receptive-field effects such as non-isotropic surround suppression.

Another question is whether the vignetting effects Roth et al. demonstrate fully explain orientation decoding. The original study by Kamitani and Tong, 2005 used a wider annular aperture reaching further into the central region, where receptive fields are smaller (inner edge: 1.5° outer edge: 10° eccentricity from fixation). The interior parts of the stimulus may therefore not be affected by vignetting. Moreover, Wardle, Ritchie, Seymour, and Carlson, 2017 showed that vignetting is not necessary for orientation decoding.

Strengths

-Well-motivated and elegant stimulus design.

-3T and 7T fMRI data from a total of 14 subjects.

-Compelling results demonstrating that vignetting can cause coarse-scale patterns that enable orientation decoding,

Weaknesses

-The paper claims to introduce a novel idea that requires reinterpretation of a large literature. The claim of novelty is unjustified. Vignetting was discovered by Carlson et al., 2014 and in Wardle et al., 2017, Carlson's group showed that it may be one, but not the only contributing factor enabling orientation decoding. Carlson et al. deserve clearer credit throughout.

-The paper doesn't attempt to address whether decoding is still possible in the absence of vignetting effects, i.e. far from the aperture boundary.

Comments and suggestions

While the experiments and analyses are excellent and the paper well written, the current version is compromised by some exaggerated claims, suggesting greater novelty and consequence than is appropriate. This should be corrected.

Abstract: "Here, we show that a large body of research that purported to measure orientation tuning may have in fact been inadvertently measuring sensitivity to second-order changes in luminance, a phenomenon we term 'vignetting'."

Abstract: "Our results demonstrate that stimulus vignetting can wholly determine the orientation selectivity of responses in visual cortex measured at a macroscopic scale, and suggest a reinterpretation of a well-established literature on orientation processing in visual cortex."

Introduction: "Our results provide a framework for reinterpreting a wide-rangeof findings in the visual system."

Too strong of a claim of novelty. The effect beautifully termed "vignetting" here was discovered by Carlson, 2014, and that study deserves the credit for triggering a reevaluation of the literature, which began three years ago. The present study does place vignetting in a broader context, discussing a variety of mechanisms by which aperture edges might influence responses, but the basic idea, including that the key factor is the interaction between the edge and the grating orientation and that the edge need not be hard, are all introduced in Carlson, 2014. The present study very elegantly demonstrates the phenomenon with fMRI, but the effect has also previously been studied with fMRI by Wardle et al., 2017, so the fMRI component doesn't justify this claim, either. Finally, while results compellingly show that vignetting was a strong contributor in Freeman et al., 2011, they don't show that it is the only contributing factor for orientation decoding. In particular, Wardle et al., 2017 suggests that vignetting in fact is not necessary for orientation decoding.

Introduction: "We and others, using fMRI, discovered a coarse-scale orientation bias in human V1; each voxel exhibits an orientation preference that depends on the region of space that it represents (Furmanski and Engel, 2000; Sasaki et al., 2006; Mannion et al., 2010; Freeman et al., 2011; Freeman et al., 2013; Larsson et al., 2017). We observed a radial bias in the peripheral representation of V1: voxels that responded to peripheral locations near the vertical meridian tended to respond most strongly to vertical orientations; voxels along the peripheral horizontal meridian responded most strongly to horizontal orientations; likewise for oblique orientations. This phenomenon had gone mostly unnoticed previously. We discovered this striking phenomenon with fMRI because fMRI covers the entire retinotopic map in visual cortex, making it an ideal method for characterizing such coarse-scale representations."

A bit too much chest thumping. The radial-bias phenomenon was discovered by Sasaki et al., 2006. Moreover, the present study negates the interpretation in Freeman et al., 2011. Freeman et al., 2011 interpreted their results as indicating an over-representation of radial orientations in cortical neurons. According to the present study, the results were in fact an artifact of vignetting and whether neuronal biases played any role is questionable. Note that Freeman et al. used a narrower and more eccentric annulus than other studies (e.g. Kamitani and Tong, 2005), so may have been more susceptible to the vignetting artifact. The authors suggest that a large literature be reinterpreted, but apparently not their own study for which they specifically and compellingly show how vignetting probably affected it.

"A leading conjecture is that the orientation preferences in fMRI measurements arise primarily from random spatial irregularities in the fine-scale columnar architecture (Boynton, 2005; Haynes and Rees, 2005; Kamitani and Tong, 2005). […] On the other hand, we have argued that the coarse-scale orientation bias is the predominant orientation-selective signal measured with fMRI, and that multivariate decoding analysis methods are successful because of it (Freeman et al., 2011; Freeman et al., 2013). This conjecture [that coarse-scale orientation bias is the predominant signal] remains controversial because the notion that fMRI is sensitive to fine-scale neural activity is highly attractive, even though it has been proven difficult to validate empirically (Alink et al., 2013; Pratte et al., 2016; Alink et al., 2017)."

This passage is a bit biased. First, the present results question the interpretation of Freeman et al., 2011. While the authors' new interpretation (following Carlson, 2014) also suggests a coarse-scale contribution, it fundamentally changes the account. Moreover, the conjecture that coarse-scale effects play a role is not controversial. What is controversial is the claim that only coarse-scale effects contribute to fMRI orientation decoding. This extreme view is controversial not because it is attractive to think that fMRI can exploit fine-grained pattern information, but because the cited studies (Alink et al., 2013, Pratte et al., 2016, Alink et al., 2017, and additional studies, including Shmuel et al., 2010) present evidence in favor of a contribution from fine-grained patterns. The way these studies are cited would suggest to an uninformed reader that they provide evidence against a contribution from fine-grained patterns. More evenhanded language is in order here.

"the model we use is highly simplified; for example, it does not take intoaccount changes in spatial frequency tuning at greater eccentricities. Yet, despite the multiple sources of noise and the simplified assumptions of the model, the correspondence between the model's prediction and the empirical measurements are highly statistically significant. From this, we conclude that stimulus vignetting is a primary source of the course scale bias."

This argument is not compelling. A terrible model may explain a portion of the explainable variance that is minuscule, yet highly statistically significant. In the absence of inferential comparisons among multiple models and model checking (or a noise ceiling), better to avoid such claims.

Discussion: "One study (Alink et al., 2017) used inner and outer circular annuli, but added additional angular edges, the result of which should be a combination of radial and tangential biases. Indeed, this study reported that voxels had a mixed pattern of selectivity, with a considerable number of voxels reliably preferring tangential gratings, and other voxels reliably favoring radial orientations."

This reasoning makes sense. The additional edges between the patches (though perhaps not well described as vignetting) complicate the interpretation of the results of Alink et al., 2011. It would be good to check the strength of the effect by simulation. Happy to share the stimuli if someone wanted to look into this.

---

## [Author Response]

Essential revisions:I have provided a summary of essential revisions. You will find more details, as well as additional points that need to be addressed, in the original reviews (appended below).1) Tone down claims of novelty:-The paper claims to introduce a novel idea that requires reinterpretation of a large literature. The claim of novelty is unjustified. Vignetting was discovered by Carlson et al., 2014 and in Wardle et al., 2017, Carlson's group showed that it may be one, but not the only contributing factor enabling orientation decoding. Carlson et al. deserve clearer credit throughout.See reviewer #2 point 1, and reviewer #3.

Carlson’s 2014 paper on stimulus edge eﬀects was an insightful contribution to the field. His work inspired much of what we did here and we have edited our manuscript in several places to cite Carlson more prominently. We would like to point out how our theoretical framework diﬀers from the work described in Carlson, 2014. Carlson used simulations to show that edge eﬀects might contribute to orientation decoding, but he did not provide a mathematical/ theoretical explanation of those edge eﬀects. This left open the possibility that Carlson’s simulation results might not have applied generally. It was even suggested that Carlson’s results were due to an artifact, a computing error, unrelated to the brain’s processing of orientation (Cliﬀord and Mannion, 2015).

We have developed a mathematical/theoretical explanation for Carlson’s ‘edge eﬀect’ based on the spread of Fourier power caused by the presence of a contrast change in the image. This is a much more general observation, which we call stimulus vignetting, and is actually much more than a simple edge artifact. We implemented an image-computable model of V1 that is capable of simulating responses to arbitrary stimuli, and we show that this model exhibits coarse-scale orientation-biases due to vignetting. The model has orientation-selective linear filters to simulate simple cells and energy filters (sum of squared responses of quadrature pairs of linear filters) for complex cells. The model was designed so that the filters evenly tile retinotopic space, orientation, and spatial frequency to avoid any possibility of the kind of artifact that was suggested to underlie Carlson’s simulation results. There are no so-called “2nd order filters” in our model. Yet it exhibits orientation-selective biases for 2nd-order modulation because of vignetting. This is an important theoretical result with widespread ramifications that go well beyond the interpretation of fMRI decoding (see below).

We tested the model’s predictions with empirical data and found a robust correspondence. Our empirical results completely contradict empirical results from Carlson’s lab (Wardle et al., 2017) in that we found measured fMRI responses could be entirely predicted by vignetting.

The eﬀect of contrast changes on Fourier content has been known in the psychophysical literature for decades (Carter and Henning, 1971). Carlson pointed out that vignetting could give rise to the coarse-scale bias observed in Freeman et al., 2011. This possibility was, in fact, raised and tested by Freeman et al., 2011. Carlson’s results are fully consistent with our conclusion in Freeman et al., 2011 that coarse-scale orientation biases, and not a fine-scale bias from column sampling, are necessary and suﬃcient for orientation decoding.

2) Provide a more nuanced coverage of the literature. E.g., the conjecture that the orientation preferences in fMRI measurements arise primarily from random spatial irregularities in the fine-scale columnar architecture doesn't seem 'leading' anymore in 2018, and more nuanced positions have been articulated since.See reviewer #1 point 1 and reviewer #3.

We have modified the text so as to describe the range of hypotheses that are currently in favor, ranging from those that believe that fMRI measurements at standard resolution are sensitive to fine-scale structure, to those who believe that fMRI at standard resolution is only sensitive to coarse-scale structure.

Our understanding of the literature is that the leading hypothesis is that orientation preferences in fMRI arise primarily from sampling of random fine-scale, columnar patterns of orientation selectivity. This hypothesis is assumed as the default explanation in several recent papers. For example, in a recent and highly cited (>200 citations in 6 years) review (Tong and Pratte, 2012), the authors state: “These orientation-decoding studies suggest that pattern analysis can be used to detect signals of columnar origin by pooling weakly feature-selective signals that can be found at the scale of millimeters, presumably due to variability in the organization of the columns.”. Reviewer 1 cites Swisher et al., 2010 as an example of a more nuanced view. The results presented in Swisher et al., 2010 do not provide evidence for the random-bias over the coarse-scale bias hypothesis. The reason is that the radial bias is broad-band, as is the polar angle component of the retinotopic map (Freeman et al., 2011). Yet, in a recent review, Tong and Pratte, 2012 write: “high-resolution functional imaging studies of the cat and human visual cortices have provided support for [the random bias] hypothesis (Swisher et al. 2010).” The random-bias hypothesis is still a leading conjecture today. We have replaced the above- mentioned sentence with the following one: “A number of previous studies have asserted that orientation preferences in fMRI measurements arise primarily from random spatial irregularities in the fine-scale columnar architecture (Boynton 2005, Haynes and Rees 2005, Kamitani and Tong, 2005)."

3) Discuss the broader implications of the study: the authors claim that the study has wide implications for many studies that used decoding of oriented gratings. But it is left unspecified what those implications are. Could the authors be more specific? For example, how should we reinterpret Kamitani and Tong, 2005 or Haynes and Rees, 2005? What wrong conclusions have been drawn, if we accept the notion that stimulus vignetting is the source of orientation decoding?The significance of the present work might be further emphasized by relating it more broadly to the general approach of MVPA (i.e. using linear classifiers to decode activity). I believe the larger lesson, highlighted by the present work, is that even seemingly simple properties are in fact hard to isolate experimentally and that powerful approaches like classification can pick up on aspects of the data that might not be what the experimenter intended.See reviewer #1 point 4, reviewer #2 point 2.

We have modified the Introduction and Discussion to address the broader implications of this study. We regret that the initial submission, unfortunately, was misleading.

The most important implication of our study is that stimulus vignetting could potentially impact any measurement of orientation selectivity, including single unit recording, calcium imaging, fMRI, and psychophysics. Most visual neuroscience experiments utilize stimuli that are restricted to a relatively small portion of the visual field. In such cases (i.e., nearly all vision experiments), vignetting can confound the interpretation of the results. The specific experimental method is irrelevant. If single-cell recordings are used to measure orientation tuning, but the receptive fields overlap the stimulus edge, or a 2nd-order modulator, then vignetting will alter the measured orientation selectivity. Vignetting, consequently, is relevant for the entire field of visual neuroscience. As we now explain in the Discussion, vignetting may underlie ostensible selectivity to second-order features such as the orientation of contrast changes (i.e. vignettes), illusory contours, and second-order texture patterns. We have now laid this out more clearly and explicitly in the Discussion section. We also suggest in the Discussion a possible reinterpretation of LGN orientation selectivity. Two recent studies, an fMRI study in humans and a calcium imaging study in mice, have claimed that orientation selectivity can be found in the LGN. However, vignetting was present in both studies. In the fMRI study the stimulus edge provided the vignette, and in the mice study the large receptive fields probably extended beyond the edge of the screen. The reinterpretation necessary, in these cases, involves considering the possibility that these studies were not necessarily measuring orientation sensitivity. Instead, the differential orientation responses could reflect solely vignetting. Therefore, it is possible that neurons in human and mouse LGN are not sensitive to orientation, as suggested originally by Hubel and Wiesel (1962).

As for fMRI orientation decoding studies, such as Kamitani and Tong, 2005 and Haynes and Rees, 2005, we have already shown that a coarse-scale bias is the dominant source of decodable orientation information (Freeman et al., 2011, 2013). In the current study we characterize vignetting as one of the sources of this coarse-scale bias. In the current study, we did not find evidence for any additional source for the coarse-scale bias, including a gain-map and/or asymmetric surround-suppression. But we cannot rule such sources out, as discussed in the Discussion section.

As now explained in the Discussion, we are no longer arguing about whether orientation decoding is dominated by coarse-scale orientation biases. We already know that it is. Note in particular the results reported by Freeman et al., 2013 in which we showed that orientation-decoding is unaffected when the slices are shifted by 1 mm (i.e., half the size of human V1 hypercolumns). We have no doubt that there are orientation columns in human V1. But we also have no doubt that orientation-decoding with fMRI at conventional resolutions is dominated by coarse-scale orientation bias, not the fine scale (columnar) architecture for orientation-selectivity. Otherwise, decoding would have suffered from shifting the slices by half the hypercolumn width. Rather than continuing to debate about whether fMRI decoding is dominated by coarse- or fine-scale information, the current study characterizes one of the sources of the coarse-scale bias, the implications of which are much broader than the technical details of orientation-decoding with fMRI (see above).

Leaving aside this debate about whether orientation decoding with fMRI depends on coarse- vs. fine-scale information, it is important to remember that the contribution of many decoding studies remains, irrespective of the source of decodable information. For example, a primary contribution of Kamitani and Tong, 2005 regards feature-based attention: they showed that attending to one or the other orientation changes the activity in V1 in such a way that the attended orientation can be decoded. That result holds regardless of whether the decoding depends on coarse- vs fine-scale. Similarly, Haynes and Rees, 2005 showed that V1 activity reflects properties of invisible stimuli, using decoding. Likewise for many studies that used decoding, forward modeling, and other MVPA techniques to reveal properties of cortical processing. Most, if not all, of these findings do not depend on the source of decodable information.

4) Address/discuss whether decoding is still possible in the absence of vignetting effects, or is solely dependent on vignetting. See reviewer #1 point 2, 3; reviewer #3.

The experiment was not designed to address this question, and we cannot definitively answer this question. It is possible — we think likely — that vignetting is the sole source of information for decoding orientation from fMRI measurements at conventional resolutions. But it is also possible that there are additional sources to the coarse-scale bias, such as a gain-map and/or asymmetric surround suppression, as discussed in the Discussion section. Our previous studies on coarse-scale biases (Freeman et al., 2011, and Freeman et al., 2013) provide more than enough evidence that orientation decoding is dominated by coarse-scale orientation biases.

Other investigators continue to disagree with this conclusion, but we have yet to see convincing evidence that fine-scale features contribute to orientation decoding at conventional fMRI resolutions, and the goal of our study was not to look for such evidence.

Can a simple experiment, consisting of full-field gratings answer this question? It may seem so, since large stimuli should evoke vignetting only at the far periphery, so that any orientation bias at mid-eccentricities would appear to be the result of neural orientation selectivity. Specifically, reviewers 1 and 3 suggest using stimuli covering a wider portion of the visual field, which should push the coarse-scale bias associated with vignetting out to the far periphery. However, such an experiment cannot, in fact, determine whether or not orientation information is available in the absence of vignetting. The reason is that, in such an experiment, one cannot rule out the possibility that voxels with population receptive fields centered at mid-eccentricities respond to the stimulus and/or screen edge. In other words, pRFs modeled as simple Gaussian functions, often have heavy tails so that they respond, albeit with small amplitude, to stimuli presented far away from their centers. Reviewer 3, in his online version of the review, suggested dividing V1 into voxels that do respond to the stimulus edge and voxels that do not. If vignetting is the source of orientation information, then an ROI of voxels that do respond to the edge should enable orientation decoding, while an ROI of voxels that do not respond to the edge should not support decoding. But the logic of the approach suggested by reviewer 3 is statistically flawed. In a standard fMRI localizer we search for voxels that respond to a stimulus, and therefore our null hypothesis is that they do not respond. One generally uses a high threshold, after multiple- comparison correction, to minimize type 1, or false positive, errors. Attempting to select voxels that do not respond to the stimulus in such an experiment amounts to accepting the null hypothesis, i.e., an elementary statistical error. Since decoding methods are tailored to exploit weak signals, sub-threshold voxels exhibiting vignetting may support decoding.

More importantly, this debate about whether decoding is still possible in the absence of vignetting entirely misses the point. Rather than continuing to debate about whether fMRI decoding is possible without coarse-scale information, the current study characterizes one of the sources of the coarse-scale bias, the implications of which are much broader than the technical details of orientation-decoding with fMRI (see above).

Reviewer #1:[…] 1) Introduction section: "A leading conjecture is that the orientation preferences in fMRI measurements arise primarily from random spatial irregularities in the fine-scale columnar architecture (Boynton, 2005; Haynes and Rees, 2005; Kamitani and Tong, 2005)."This conjecture was perhaps leading in 2005, but I don't think this is an accurate description of the state of affairs 13 years later, and it therefore seems a bit of a straw man. For example, a more nuanced view articulated by Swisher et al., 2010 from the Tong lab states that orientation information "can be found at spatial scales ranging from the size of individual columns to about a centimeter". I suggest the authors paint a more balanced picture in the introduction.

See Essential revisions #2.

2) Subsection “Coarse-scale bias and stimulus vignetting: fMRI experiments”: "From this, we conclude that stimulus vignetting is a primary source of the coarse scale bias."Why “a primary source”, rather than “a source”, or 'an important source'? It seems bold to conclude this based on a correlation of ~0.2-0.26 between the model and the data?

We have removed this sentence from the text.

3) In the same section: "Decoding accuracy for the shifted between-modulator analysis was only slightly lower than within-modulator decoding accuracy".Could the authors test whether this difference is statistically significant?If the difference is significant, this could, as the authors point out, be due to the inner and outer radial edges. However, it could potentially also be caused by the fact that there is a small amount of orientation information present in the fMRI activity patterns that is not due to stimulus vignetting. The authors may also want to include this possibility in the text – even though they may find it unlikely.

We now include this possibility in the text: “However, we cannot rule out the possibility that some other orientation information, that is not dependent on the vignette, is present in the data.” (The difference is significant, according to the phase permutation test.)

4) Discussion section: "Our results provide a framework for reinterpreting a wide-range of findings on orientation selectivity, measured with both fMRI in human subjects and in single units."This statement suggests that the study has wide implications for many studies that used decoding of oriented gratings. But it is left unspecified what those implications are. Could the authors be more specific here? For example, how should we reinterpret Kamitani and Tong, 2005 or Haynes and Rees, 2005? What wrong conclusions have been drawn, if we accept the notion that stimulus vignetting is the source of orientation decoding?

See Essential revisions #3.

Reviewer #2:[…] 1) Wording and framing.- There are a few places where I think the conclusions and claims should be toned down. For example, "vast number of previous studies" (Introduction section) and "wide-range of findings in the visual system". I assume the authors are referring to past studies that have used orientation stimuli in fMRI, and not the neurophysiology literature on orientation tuning in single neurons. While it is theoretically possible that vignetting effects may be influencing single-neuron response properties (since Gabor and grating patches are widely used as stimuli), it is not yet clear whether single-neuron studies need to be re-interpreted.

See Essential revisions #3. Actually, we are referring to the literature on orientation tuning in single neurons, in addition to other methods, including fMRI. We agree that further empirical demonstrations of vignetting, using various measurement methods, would be helpful. However, based on our theoretical, computational, and fMRI results, we believe it is quite clear that vignetting may have affected many measurements of orientation selectivity, at various spatial scales including single-unit recordings.

- The work of Carlson, 2014 involves modeling work that is similar to what is done in the current manuscript. Of course, the major advance of the present work is the demonstration of empirical findings, but this previous work might deserve more acknowledgment.

We agreed with the reviewer’s comment regarding our empirical findings. See Essential revisions #1 for clarification about our views regarding the novelty of the mathematical/ theoretical framework that we describe in the manuscript.

- Ultimately, a lot of the controversy regarding fMRI orientation decoding comes down to numbers, and it would be helpful to clarify what is meant by "coarse" and "fine". I assume the authors mean something to the effect that "coarse" is > 1 mm and "fine" is < 0.5 mm (or something like that).

The precise distinction between coarse and fine orientation biases has no bearing on our results or conclusions. Surely, any distinction we choose would be challenged by one researcher or another. See Essential revisions #3.

2) Some big-picture perspective.- The significance of the present work might be further emphasized by relating it more broadly to the general approach of MVPA (i.e. using linear classifiers to decode activity). I believe the larger lesson, highlighted by the present work, is that even seemingly simple properties are in fact hard to isolate experimentally and that powerful approaches like classification can pick up on aspects of the data that might not be what the experimenter intended. One way that we have conceptualized this (Naselaris TICS, 2015) is that the orientation of a grating stimulus is not the only stimulus feature that can give rise to variance in data, and that classification can reflect a number of different stimulus features, such as those related to vignette effects, unless one does work to rule them out, e.g. by considering explicit computational models.3) Acknowledgment of the limitations of the model.- I think the main contribution of the current paper is the experimental results. The modeling analyses do provide value in that they demonstrate a concrete (and reasonably plausible) explanation of what could be driving the observed orientation-tuning results. However, as the authors recognize in the Discussion, there are many stimulus properties beyond what is characterized in the model that are known to affect V1 responses (e.g. surround suppression, contour integration, 2nd-order contrast effects, etc.), and which might also contribute to the orientation effects. Thus, the text could be clarified to acknowledge the limitations of the model and indicate what role the modeling results play in this specific paper. It seems that the role of the modeling results is to show concretely that imbalances in filter energy across orientations exist at stimulus edges and that this is one possible reason for finding orientation tuning in standard-resolution fMRI. (Note that I am not suggesting that the present paper needs to perform detailed model comparisons (in which different models are pitted to quantitatively account for individual voxel responses to a variety of stimulus conditions); that would be outside of the scope of this paper.)

We added a paragraph to the Discussion to clarify the purpose of the model. The purpose of the model was to: 1) demonstrate that stimulus vignetting is sufficient to create a radial bias, 2) create novel stimuli with predicted opposite biases, and 3) provide predictions of single voxel responses in response to the novel stimuli; the model is image-computable and accepts arbitrary stimuli as input. The strength of the model is its simplicity. Our goal is not to provide an accurate and complete model of V1 activity, but rather to provide a platform for assessing vignetting. We purposely kept it simple so that we could be certain that vignetting is indeed the source of orientation bias in the model. The point is that ostensibly interesting features of V1 responses might actually be a result of vignetting. For example, sensitivity of V1 neurons to second-order changes might actually reflect vignetting. If we were to include asymmetric surround suppression in the model, any orientation bias could be a combined result of vignetting and surround suppression.

4) Clarification of the modeled effect- It would be helpful to isolate and clarify the nature of the effect shown in Figure 2. One potential explanation is that for a linear filter stimulated with an optimal oriented grating, the filter shows a bigger response when the grating has a vignette edge orthogonal to the orientation compared to when the grating has a vignette edge parallel to the orientation. Is this the case in the model?

A linear filter will not necessarily respond more strongly to an orthogonal vignette vs. a parallel vignette. That would depend on the specific neuron’s receptive field. For example, in Figure 1, the neuron with a receptive field depicted in green would respond more to the vertical edge than to the horizontal edge, while the red neuron would respond more strongly to the horizontal edge.

Reviewer #3:[…] The study is beautifully designed and expertly executed. Results compellingly demonstrate that, as proposed by Carlson, 2014, vignetting can account for the coarse-scale biases reported in Freeman et al., 2011. The paper also contains a careful discussion that places the phenomenon in a broader context. Vignetting describes a family of effects related to aperture edges and their interaction with the contents of the aperture. The interaction could be as simple as the aperture edge adding edge energy of a different orientation and thus changing orientation selective response. It could also involve extra-receptive-field effects such as non-isotropic surround suppression.*Another question is whether the vignetting effects Roth et al. demonstrate fully explain orientation decoding. The original study by Kamitani and Tong, 2005 used a wider annular aperture reaching further into the central region, where receptive fields are smaller (inner edge: 1.5*° *outer edge: 10*° eccentricity from fixation). The interior parts of the stimulus may therefore not be affected by vignetting. Moreover, Wardle, Ritchie, Seymour, and Carlson, 2017 showed that vignetting is not necessary for orientation decoding.

See Essential revisions #4

Strengths-Well-motivated and elegant stimulus design.-3T and 7T fMRI data from a total of 14 subjects.-Compelling results demonstrating that vignetting can cause coarse-scale patterns that enable orientation decoding.Weaknesses-The paper claims to introduce a novel idea that requires reinterpretation of a large literature. The claim of novelty is unjustified. Vignetting was discovered by Carlson et al., 2014 and in Wardle et al., 2017, Carlson's group showed that it may be one, but not the only contributing factor enabling orientation decoding. Carlson et al. deserve clearer credit throughout.

See Essential revisions #1

-The paper doesn't attempt to address whether decoding is still possible in the absence of vignetting effects, i.e. far from the aperture boundary.

See Essential revisions #4

Comments and suggestionsWhile the experiments and analyses are excellent and the paper well written, the current version is compromised by some exaggerated claims, suggesting greater novelty and consequence than is appropriate. This should be corrected.Abstract: "Here, we show that a large body of research that purported to measure orientation tuning may have in fact been inadvertently measuring sensitivity to second-order changes in luminance, a phenomenon we term 'vignetting'."Abstract: "Our results demonstrate that stimulus vignetting can wholly determine the orientation selectivity of responses in visual cortex measured at a macroscopic scale, and suggest a reinterpretation of a well-established literature on orientation processing in visual cortex."Introduction: "Our results provide a framework for reinterpreting a wide-rangeof findings in the visual system."

We stand by these statements.

Too strong of a claim of novelty. The effect beautifully termed "vignetting" here was discovered by Carlson, 2014, and that study deserves the credit for triggering a reevaluation of the literature, which began three years ago. The present study does place vignetting in a broader context, discussing a variety of mechanisms by which aperture edges might influence responses, but the basic idea, including that the key factor is the interaction between the edge and the grating orientation and that the edge need not be hard, are all introduced in Carlson, 2014. The present study very elegantly demonstrates the phenomenon with fMRI, but the effect has also previously been studied with fMRI by Wardle et al., 2017, so the fMRI component doesn't justify this claim, either. Finally, while results compellingly show that vignetting was a strong contributor in Freeman et al., 2011, they don't show that it is the only contributing factor for orientation decoding. In particular, Wardle et al., 2017 suggests that vignetting in fact is not necessary for orientation decoding.

See Essential revision #1.

Indeed, our results contradict those of Wardle et al., 2017.

We are not sure which “claim of novelty” the reviewer is referring to. All excerpts seem accurate to us.

Introduction: "We and others, using fMRI, discovered a coarse-scale orientation bias in human V1; each voxel exhibits an orientation preference that depends on the region of space that it represents (Furmanski and Engel, 2000; Sasaki et al., 2006; Mannion et al., 2010; Freeman et al., 2011; Freeman et al., 2013; Larsson et al., 2017).[…] This phenomenon had gone mostly unnoticed previously. We discovered this striking phenomenon with fMRI because fMRI covers the entire retinotopic map in visual cortex, making it an ideal method for characterizing such coarse-scale representations."A bit too much chest thumping. The radial-bias phenomenon was discovered by Sasaki et al., 2006. Moreover, the present study negates the interpretation in Freeman et al., 2011. Freeman et al., 2011 interpreted their results as indicating an over-representation of radial orientations in cortical neurons. According to the present study, the results were in fact an artifact of vignetting and whether neuronal biases played any role is questionable. Note that Freeman et al. used a narrower and more eccentric annulus than other studies (e.g. Kamitani and Tong, 2005), so may have been more susceptible to the vignetting artifact. The authors suggest that a large literature be reinterpreted, but apparently not their own study for which they specifically and compellingly show how vignetting probably affected it.

We cite Sasaki, 2006 along with an earlier paper by Furmanski and Engel, 2000, in the opening sentence of the second paragraph of the Introduction, along with our own papers. We do not prioritize our own work over these other papers.

We completely agree that our previous papers, including Freeman et al., 2011 were susceptible to vignetting. In fact, our claim in the current paper is that the results reported in our previous papers were dominated by vignetting. It is true that Freeman et al. suggested that the coarse- scale bias may reflect a gain-map. However, this suggestion was presented as just that – a suggestion that was made in the Discussion section of the 2011 paper. Indeed, in our current study we found no evidence for a gain-map, but as we state in the Discussion section of the current manuscript, we cannot (based on the current results) rule out additional sources of the coarse-scale bias, including a gain-map. In fact, the reviewer seems to be aware of this (“while results compellingly show that vignetting was a strong contributor in Freeman et al., 2011, they don't show that it is the only contributing factor”, two paragraphs above).

"A leading conjecture is that the orientation preferences in fMRI measurements arise primarily from random spatial irregularities in the fine-scale columnar architecture (Boynton, 2005; Haynes and Rees, 2005; Kamitani and Tong, 2005). […] On the other hand, we have argued that the coarse-scale orientation bias is the predominant orientation-selective signal measured with fMRI, and that multivariate decoding analysis methods are successful because of it (Freeman et al., 2011; Freeman et al., 2013). This conjecture [that coarse-scale orientation bias is the predominant signal] remains controversial because the notion that fMRI is sensitive to fine-scale neural activity is highly attractive, even though it has been proven difficult to validate empirically (Alink et al., 2013; Pratte et al., 2016; Alink et al., 2017)."This passage is a bit biased. First, the present results question the interpretation of Freeman et al., 2011. While the authors' new interpretation (following Carlson, 2014) also suggests a coarse-scale contribution, it fundamentally changes the account. Moreover, the conjecture that coarse-scale effects play a role is not controversial. What is controversial is the claim that only coarse-scale effects contribute to fMRI orientation decoding. This extreme view is controversial not because it is attractive to think that fMRI can exploit fine-grained pattern information, but because the cited studies (Alink et al., 2013, Pratte et al., 2016, Alink et al., 2017, and additional studies, including Shmuel et al., 2010) present evidence in favor of a contribution from fine-grained patterns. The way these studies are cited would suggest to an uninformed reader that they provide evidence against a contribution from fine-grained patterns. More evenhanded language is in order here.

We have changed the text to read: “However, the finding that a coarse-scale bias is the source of orientation decoding remains controversial, and several recent studies have attempted to disprove it (Alink, Krugliak et al., 2013, Pratte, Sy et al., 2016, Alink, Walther et al., 2017), in part, we believe, because the notion that fMRI is sensitive to fine-scale neural activity is highly attractive.”

"the model we use is highly simplified; for example, it does not take intoaccount changes in spatial frequency tuning at greater eccentricities. Yet, despite the multiple sources of noise and the simplified assumptions of the model, the correspondence between the model's prediction and the empirical measurements are highly statistically significant. From this, we conclude that stimulus vignetting is a primary source of the course scale bias."This argument is not compelling. A terrible model may explain a portion of the explainable variance that is minuscule, yet highly statistically significant. In the absence of inferential comparisons among multiple models and model checking (or a noise ceiling), better to avoid such claims.

We have removed this sentence from the text.

Discussion: "One study (Alink et al., 2017) used inner and outer circular annuli, but added additional angular edges, the result of which should be a combination of radial and tangential biases. Indeed, this study reported that voxels had a mixed pattern of selectivity, with a considerable number of voxels reliably preferring tangential gratings, and other voxels reliably favoring radial orientations."This reasoning makes sense. The additional edges between the patches (though perhaps not well described as vignetting) complicate the interpretation of the results of Alink et al., 2011. It would be good to check the strength of the effect by simulation. Happy to share the stimuli if someone wanted to look into this.

We intend to share publicly the code for our model so that the reviewer (or anyone else) can follow up.